# Rolling circle RNA synthesis catalyzed by RNA

Emil Laust Kristoffersen[1†], Matthew Burman[2], Agnes Noy[2], Philipp Holliger[1]*

[1]MRC Laboratory of Molecular Biology, Cambridge Biomedical Campus, Cambridge, United Kingdom; [2]Department of Physics, University of York, York, United Kingdom

**Abstract** RNA-catalyzed RNA replication is widely considered a key step in the emergence of life's first genetic system. However, RNA replication can be impeded by the extraordinary stability of duplex RNA products, which must be dissociated for re-initiation of the next replication cycle. Here, we have explored rolling circle synthesis (RCS) as a potential solution to this strand separation problem. We observe sustained RCS by a triplet polymerase ribozyme beyond full-length circle synthesis with strand displacement yielding concatemeric RNA products. Furthermore, we show RCS of a circular Hammerhead ribozyme capable of self-cleavage and re-circularization. Thus, all steps of a viroid-like RNA replication pathway can be catalyzed by RNA alone. Finally, we explore potential RCS mechanisms by molecular dynamics simulations, which indicate a progressive build-up of conformational strain upon RCS with destabilization of nascent strand 5'- and 3'-ends. Our results have implications for the emergence of RNA replication and for understanding the potential of RNA to support complex genetic processes.

## Editor's evaluation

This paper is of interest to scientists from the field of origin of life or RNA synthesis in general, especially those interested in the "RNA world" scenario. The data analysis is rigorous and the conclusions are justified by the data. The key claims of the manuscript are directly related to, and support, previous findings.

*For correspondence:
ph1@mrc-lmb.cam.ac.uk

Present address:
†Interdisciplinary Nanoscience Center (iNANO), Aarhus University, Aarhus, Denmark

Competing interest: The authors declare that no competing interests exist.

## Introduction

The versatility of RNA functions underpins hypotheses regarding the origin and early evolution of life. Such hypotheses of an 'RNA world'—a primordial biology centered on RNA as the main biomolecule—are in accord with the essential role of RNA catalysis in present-day biology (*Cech, 2000*; *Goldman and Kacar, 2021*; *Nissen et al., 2000*; *Wilkinson et al., 2020*) and the discovery of multiple prebiotic synthetic pathways to several of the RNA (and DNA) nucleotides (*Becker et al., 2019*; *Kim et al., 2020*; *Patel et al., 2015*; *Powner et al., 2009*; *Xu et al., 2020*). In addition, progress in both non-enzymatic (*Deck et al., 2011*; *Hassenkam et al., 2020*; *Prywes et al., 2016*; *Rajamani et al., 2008*; *Sosson et al., 2019*; *Sponer et al., 2021*; *Wachowius and Holliger, 2019*; *Zhang et al., 2020*; *Zhou et al., 2020*) and RNA-catalyzed polymerization of RNA and some of its analogs (*Attwater et al., 2018*; *Attwater et al., 2013*; *Cojocaru and Unrau, 2021*; *Ekland and Bartel, 1996*; *Horning and Joyce, 2016*; *Johnston et al., 2001*; *Mutschler et al., 2018*; *Shechner et al., 2009*; *Tagami et al., 2017*; *Tjhung et al., 2020*) is beginning to map out a plausible path to RNA self-replication; a cornerstone of the RNA world hypothesis.

RNA in vitro evolution and engineering have led to the discovery of RNA polymerase ribozymes (RPRs) able to perform templated RNA synthesis of up to ~200 nucleotides (nt) (*Attwater et al., 2013*), synthesizing active ribozymes including the catalytic class I ligase core (*Horning and Joyce,*

**eLife digest** Many organisms today rely on a trio of molecules for their survival: DNA, to store their genetic information; proteins, to conduct the biological processes required for growth or replication; and RNA, to mainly act as an intermediary between DNA and proteins. Yet, how these inanimate molecules first came together to form a living system remains unclear.

Circumstantial evidence suggests that the first lifeforms relied to a much greater exrtent on RNA to conduct all necessary biological processes. There is no trace of this 'RNA world' today, but molecular 'fossils' may exist in current biology. Viroids, for example, are agents which can infect and replicate inside plant cells. They are formed of nothing but a circular strand of RNA that serves not only as genetic storage but also as ribozymes (RNA-based enzymes). Viroids need proteins from the host plant to replicate, but scientists have been able to engineer ribozymes that can copy complex RNA strands. This suggests that viroid-like replication could be achieved using only RNA.

Kristoffersen et al. put this idea to the test and showed that it is possible to use RNA enzymatic activity alone to carry out all the steps of a viroid-like copying mechanism. This process included copying a viroid-like RNA circle with RNA, followed by trimming the copy to the right size and reforming the circle. These two latter steps could be carried out by a ribozyme that could itself be encoded on the RNA circle. A computer simulation indicated that RNA synthesis on the circle caused increasing tension that could ease some of the barriers to replication.

These results increase our understanding of how RNA copying by RNA could be possible. This may lead to developing molecular models of a primordial RNA-based replication, which could be used to investigate early genetic systems and may have potential applications in synthetic biology.

*2016*; *Tjhung et al., 2020*) at the heart of the most efficient RPRs, as well as initiate processive RNA synthesis using a mechanism with analogies to sigma-dependent transcription initiation (*Cojocaru and Unrau, 2021*). An RPR capable of utilizing trinucleotide triphosphates (triplets) as substrates (a triplet polymerase ribozyme [TPR]) has been shown to display an enhanced capacity to copy highly structured RNA templates including segments of its own sequence (*Attwater et al., 2018*).

Nevertheless, there remain a number of fundamental obstacles to be overcome before an autonomous self-replication system can be established. A central problem among these is the so-called 'strand separation problem,' a form of product inhibition due to the accumulation of highly stable dead-end RNA duplexes, which cannot be dissociated (efficiently) under replication conditions (*Le Vay and Mutschler, 2019*). The strand separation problem has been overcome by PCR-like thermocycling (or thermophoresis) (*Horning and Joyce, 2016*; *Salditt et al., 2020*), but this approach may be limited to short RNA oligomers (even in the presence of high concentrations of denaturing agents) as the melting temperatures of longer RNA duplexes approach or even exceed the boiling point of water (*Freier et al., 1986*; *Szostak, 2012*).

While RNA duplexes occur by necessity as intermediates of RNA replication, the extent of the strand separation problem can be modulated by genome topology. Circular rather than linear genomes are widespread in biology including eukaryotes, prokaryotes, and viruses (*Moller et al., 2018*; *Moss et al., 2020*; *Shulman and Davidson, 2017*). Circular RNAs (circRNAs) are found as products of RNA splicing (*Kristensen et al., 2019*) and RNA-based self-circularization is known in multiple ribozymes (*Hieronymus and Müller, 2019*; *Lasda and Parker, 2014*; *Petkovic and Müller, 2015*). Continuous templated RNA synthesis on circular templates (rolling circle synthesis [RCS]) is also widespread and found in the replication of the RNA genomes in some viruses and in viroids. Indeed, viroid RNA replication has been proposed to resemble an ancient mechanism for replication (*Diener, 1989*; *Flores et al., 2014*).

In an idealized RCS mechanism, both strand invasion and displacement processes are isoenergetic and coordinated to nascent strand extension (*Blanco et al., 1989*; *Daubendiek et al., 1995*), with rotation of the single-stranded RNA (ssRNA) alleviating the build-up of topological tension (*Kuhn et al., 2002*). Thus, RCS is a potentially open-ended process leading to the synthesis of single-stranded multiple repeat products (concatemers) with an internally energized strand displacement (*Tupper and Higgs, 2021*). RCS as a replication mode has therefore potentially unique properties with regards to the strand separation problem. Specifically in the context of triplet-based RNA replication on a

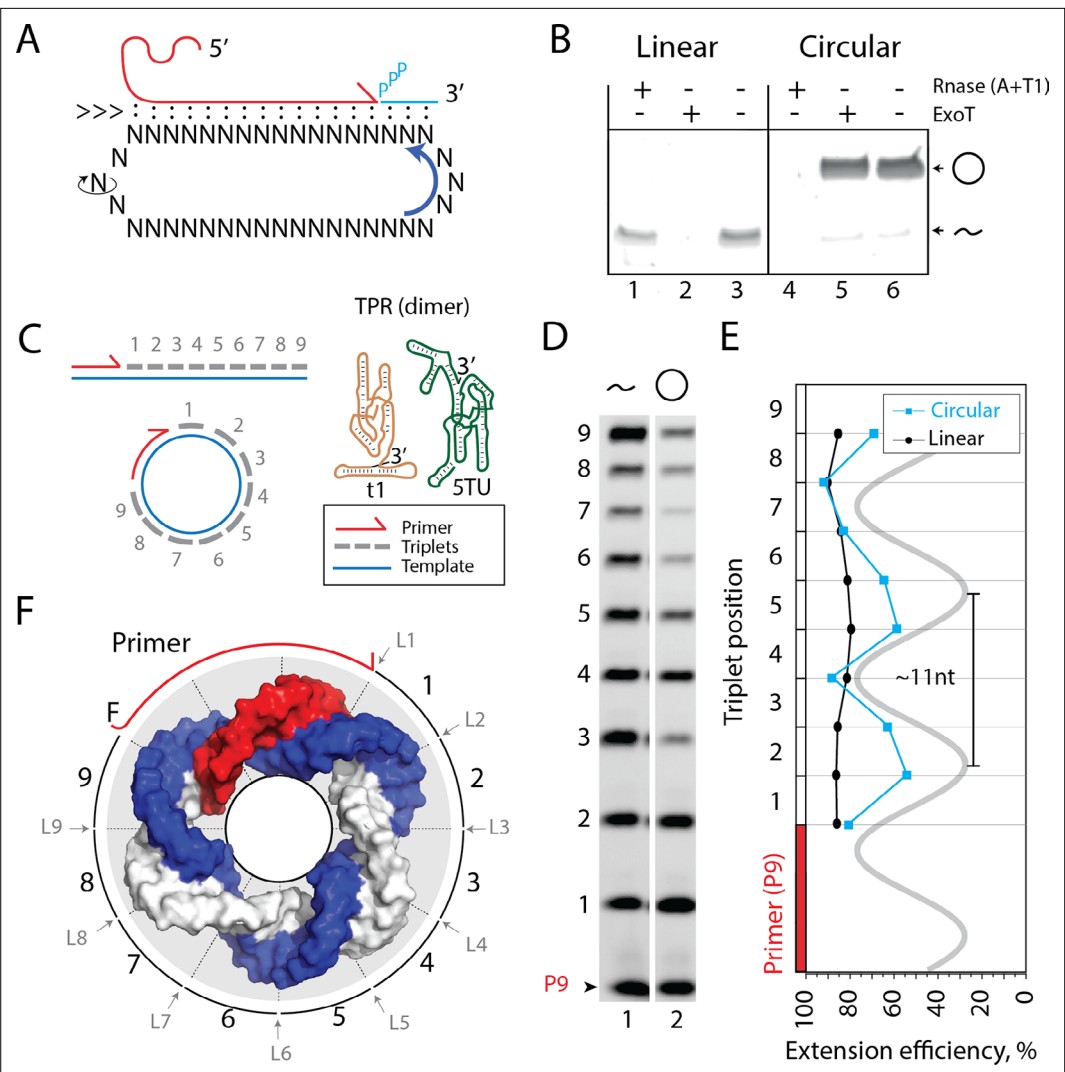

**Figure 1.** Primer extension on circular RNA (circRNA) templates. (**A**) Schematic illustration of rolling circle synthesis (RCS). Red product RNA strand is extended by a triplet at the 3'-end while three base pairs dissociate at the 5'-end keeping the total hybridization energy constant. Topological relaxation is allowed by rotation of the single stranded part of the circular template (swiveling arrow). (**B**) Linear or circularized RNA is treated with or without endo- or exonucleases (RNase A/T1 mix or Exonuclease T (ExoT)). Only circRNA is resistant to ExoT digestion. (**C**) Schematic representation of primer triplet extension on a linear or circRNA template and of the TPR hetero-dimer comprising the catalytic subunit (5TU (green)) and the non-catalytic subunit (t1 (red)) (**Attwater et al., 2018**) (TPR sequence: **Supplementary file 1**). (**D**) PAGE gel of TPR primer extension, P9 (5'-FAM-GAAGAAGAA) is the unextended primer, bands 1–9 denote extension of P9 by +1 to + 9 triplets (full length). RNA template used was sc12GAA-p (36 nt, 12 repeat UUC template). Experiment was done under standard conditions in the Tris buffer system described in Materials and Methods except that 800 pmol triplets were used. (**E**) Extension efficiency of formation of bands 1–9 in (**D**) (see Materials and methods) is plotted against triplet position. (**F**) Schematic model of the sc12GAA-p illustrating the different accessibility of in- or outside facing triplet junctions on the scRNA template (blue), with P9 primer (red), and the product strand (light gray). Original gel images and numerical values are supplied in **Figure 1—source data 1**.

The online version of this article includes the following source data and figure supplement(s) for figure 1:

**Source data 1.** Gel images and numerical values.

**Figure supplement 1.** Purification of circularized RNA (scGAA8-16).

**Figure supplement 1—source data 1.** Gel images.

circular template, duplex dissociation, and strand separation may in principle be driven by trinucleotide (triplet) hybridization and ligation, leading to extension of the nascent strand 3′-end and an equal displacement of the 5′-end in triplet increments (*Figure 1A*). Triplet binding to the template strand and dissociation of an equal trinucleotide stretch from the 5′-end are both equilibrium processes and nearly isoenergetic. However, extension (i.e., ligation of the bound triplet to the growing 3′-end) is an irreversible step. Thus, in this scenario, RCS would be expected to proceed in ratchet-like fashion with strand displacement driven by triphosphate hydrolysis and triplet ligation.

Here, we have explored triplet-based RCS of scRNA templates as a potential solution to the strand separation problem in RNA-catalyzed RNA replication. We show that RCS can be catalyzed by the triplet polymerase ribozyme [TPR] (*Attwater et al., 2018*). We show that the TPR is able to perform continuous templated extension of circular RNA templates beyond full-length circle synthesis with strand displacement yielding concatemeric RNA products. We also investigated the mechanistic basis for RCS and strand displacement by molecular dynamics (MD) simulations of scRNA in explicit solvent. Finally, we consider the potential of a full viroid-like replication cycle catalyzed by RNA alone by design and synthesis of a circular Hammerhead ribozyme capable of both product cleavage and self-circularization.

## Results
### RNA-catalyzed primer extension using small circular RNA templates
We first set out to investigate whether templated RNA synthesis on scRNAs as templates could be catalyzed by an RNA catalyst. To extend the RNA nascent strand beyond the full circle and initiate RCS requires duplex invasion and displacement of the original RNA primer and product strand. However, most RPRs are inhibited by duplex RNA both in the form of template secondary structures and as linear duplex RNA. We therefore explored the potential of a recently described TPR (*Attwater et al., 2018*), which is able to utilize trinucleotide triphosphates (triplets (ᵖᵖᵖNNN)) as substrates for polymerization. Due to increased binding of the triplets to the template (compared, e.g., to the canonical mononucleotide triphosphates (ᵖᵖᵖN, NTPs)), triplets are able to invade and cooperatively 'open up' template secondary structures for replication (*Attwater et al., 2018*). We hypothesized that this ability might also promote the continuous invasion and displacement of the nascent strand 5′-end and facilitate the RCS mechanism (*Figure 1A*).

As described previously, RNA synthesis by the TPR is most efficient in the eutectic phase of water ice, due to its beneficial reaction conditions for ribozyme catalysis (*Attwater et al., 2018*). Specifically, eutectic ice phases aid TPR activity by the reduced degree of RNA hydrolysis under low-temperature conditions, reduced water activity, and the high concentrations of reactants (ribozyme, scRNA template, triplet substrates, and Mg²⁺ ions) present in the eutectic brine phase that arises by excluding solutes from growing water ice crystals and remains liquid at subzero temperatures (*Attwater et al., 2010*). Thus, all RCS experiments were carried out under eutectic conditions.

We prepared scRNA templates (34–58 nt in length) by in vitro transcription and ligation. Circularity of the scRNA templates was verified based on gel mobility and resistance to exonuclease (exoT) degradation in contrast to the linear, non-ligated counterparts (*Figure 1B*, *Figure 1—figure supplement 1*/sequences for all oligonucleotides in *Supplementary file 1*). We first investigated primer extensions using a single triplet (ᵖᵖᵖGAA) on the scRNA template as this provides an even banding pattern of incorporation. This facilitates analysis and allows primer extension efficiencies of linear and circular templates to be more readily compared (*Figure 1C*). Primer extension experiments using a purified 36 nt scRNA as template resulted in full-length extension around the circle (*Figure 1D*), but with reduced efficiency compared to a linear RNA template. Furthermore, we observed a periodic banding pattern of triplet extension efficiency matching the helical pitch of double-stranded RNA (dsRNA) (11.3 base pairs (bp)/turn *Bhattacharyya et al., 1990*; *Figure 1E*). Presumably, triplet junctions located on the inside of the scRNA ring are less easily accessible and therefore less efficiently ligated than in linear RNA, which is freely accessible from all sides (*Figure 1F*). This may explain both the observed periodicity and reduced synthetic efficiency of RNA synthesis by the TPR on scRNAs.

Despite the reduced extension efficiency in scRNA, we obtained full circle extension products for multiple templates (34–58 nt in size, *Figure 2A and B*) with a clear trend toward increasing mean extension efficiency for circular templates with increasing size predicting parity with the linear template at

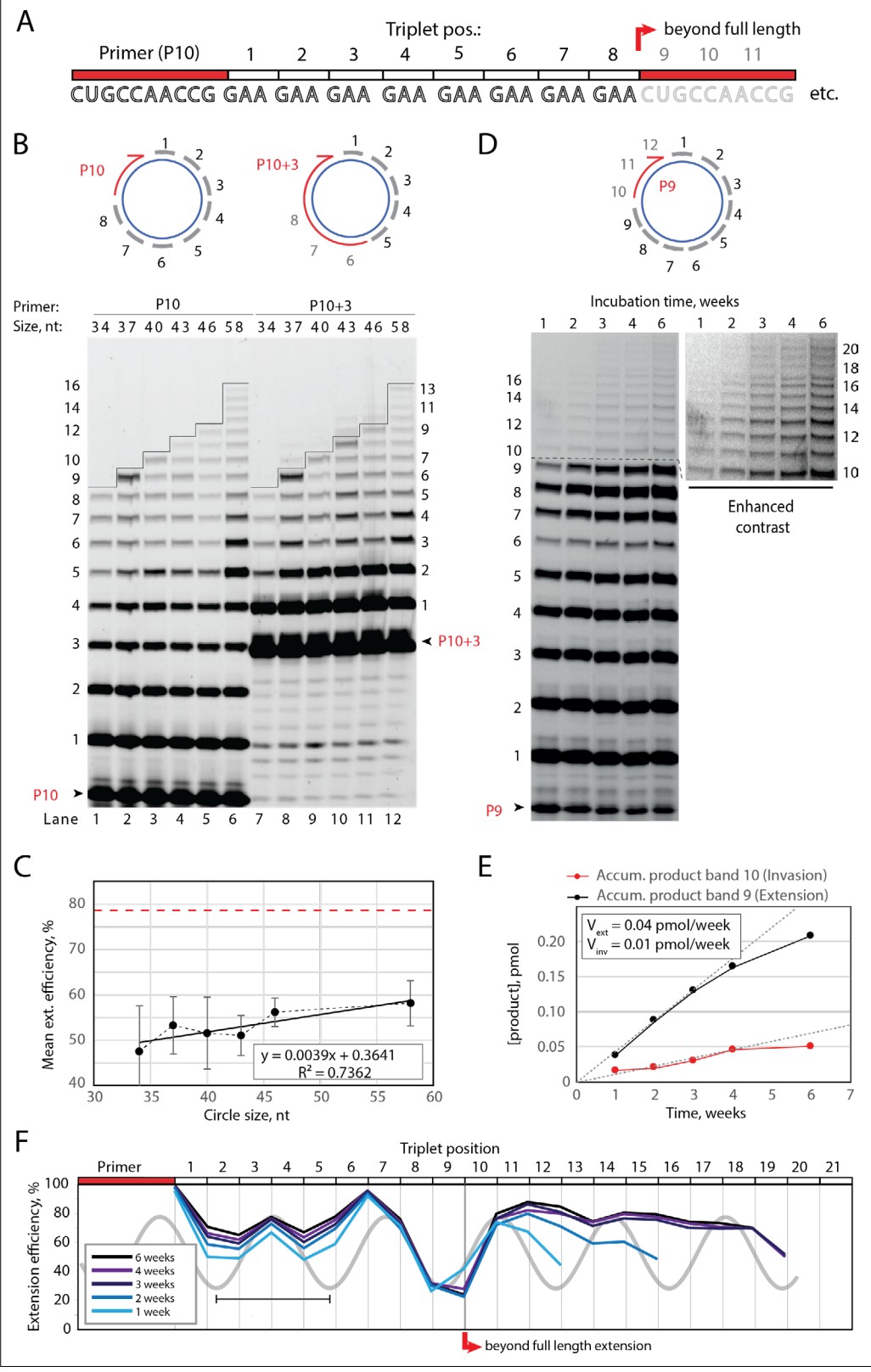

**Figure 2.** Full-length and beyond full-length RNA-catalyzed RNA synthesis on circular RNA templates. (**A**) Product strand of primer extension experiments with primer P10 (red) and eight triplet scRNA template strands. Potential beyond full-length circle synthesis triplets are shaded opaque. (**B**) Various scRNA template sizes allow full-length primer extension as indicated (with eight triplet sites) (gray), GAA triplets (black), and primers P10

*Figure 2 continued on next page*

*Figure 2 continued*

(5′-FAM-CUGCCAACCG) or P10 +3 (5′-FAM-GAAGAAGAA-CUGCCAACCG) (red). PAGE of primer extensions (under standard conditions) with full-length synthesis for different scRNA templates (34–58, termed scGAA8-16, *Supplementary file 1*) marked by a black line. (**C**) Mean extension efficiency plotted as a function of scRNA size calculated from extension experiments including (**B**) (error bars indicate standard deviation, n=5), with mean extension efficiency for a linear RNA template (red dashed line). (**D**) PAGE of time-course of primer extension of primer P9 on scRNA template, sc12GAA-p (optimized conditions). Full-length circle synthesis is marked by a dashed black line (after +9). Bands +10 and beyond (see Enhanced contrast section) indicate beyond full-length synthesis. (**E, F**) Mean extension efficiency (from (**D**)) plotted against time (**E**) or triplet position (**F**), showing the respective amounts of product at full-length (black) and beyond full-length circle (red) synthesis as well as the drop in efficiency at full length, which recovers once beyond full-length synthesis is initiated. $V_{ext}$ and $V_{inv}$ denote the calculated velocity of formation of bands 9 and 10, respectively. Original gel images and numerical values are supplied in *Figure 2—source data 1*.

The online version of this article includes the following source data and figure supplement(s) for figure 2:

**Source data 1.** Gel images and numeric values.

**Figure supplement 1.** Optimization of rolling circle synthesis.

**Figure supplement 1—source data 1.** Gel images and numeric values.

around 120 nt (*Figure 2C*). Note, in these experiments extension beyond full circle was not intended or possible (lanes 1–6 in *Figure 2B*) as the specific triplet substrates needed for displacing the primer were not present in the reaction.

Having established full-length synthesis on scRNA templates, we next tested if primer extension could proceed beyond full circle requiring duplex invasion and displacement of the primer/product strand. We first tested this using primer P10 +3, comprising a 5′ extension of three GAA repeats, thus covering the last three UUC triplet binding sites on the circular template (*Figure 2B*, top right). We observed an extension of up to +3 triplets (+9) above the full circle mark (*Figure 2B*, lanes 7–12), indicating displacement of the primer 5′-end upon incorporation of three additional ᵖᵖᵖGAA triplets. This indicated that synthesis beyond full circle including strand displacement is possible on scRNA templates, boding well for the implementation of RCS. To that effect, we next optimized buffer and extension conditions for more efficient extension above the full circle mark (*Figure 2—figure supplement 1*). Interestingly, greater dilution of reaction mixtures prior to freezing resulted in more efficient stand displacement. While greater pre-freezing dilution does not alter the final solute concentrations within the eutectic phase (*Attwater et al., 2010*), it increases the eutectic phase/ice interface area. This suggests that strand invasion may be aided by surface effects, as previously suggested for RNA refolding (*Mutschler et al., 2015*). Under these optimized buffer and extension conditions, we observed progressive accumulation of progressively longer RCS products, over prolonged reaction times (up to 6 weeks) (*Figure 2D*) with reaction speed decreasing after ca. 4 weeks incubation, indicating continued RCS over extended periods of time (*Figure 2E and F*).

## Molecular dynamics simulations of 36 nt scRNA

To better understand the structural and topological constraints of RCS on scRNAs, we performed atomistic MD simulations over 400 ns of the different RCS stages, comprising the starting scRNA template as circular ssRNA and scRNA with a progressively extended dsRNA segments (*Figure 3*). For simplicity, a 36 nt circular RNA sequence of (UUC)₁₂ was chosen as a template strand (similar to the scRNA template [sc12GAA-p] in *Figures 1 and 2D*) for direct comparison with the experimental system. The complementary strand comprising GAA triplets starting from 9 bp dsRNA (corresponding to binding of primer P9) was extended from 18 to 30 bp (after +3 triplet incorporation) in triplet increments of dsRNA corresponding to extension products of bands +3, +4, +5, +6, and +7 (see gel in *Figure 1D*), using the most representative structure of the previous simulation as a starting point for the next one.

The simulation trajectories revealed the high energy barrier of dsRNA for bending and accommodating a circular shape (*Figure 3A*). Instead, we observe that, as dsRNA is elongated, the remaining ssRNA segment of the scRNA becomes increasingly extended. As the dsRNA part reaches 27 bp (corresponding to band 6 in *Figure 1D*), the ssRNA segment was fully extended and torsional strain was relieved by dissociation (peeling off) of the dsRNA 5′- and 3′-ends rather than by bending or the

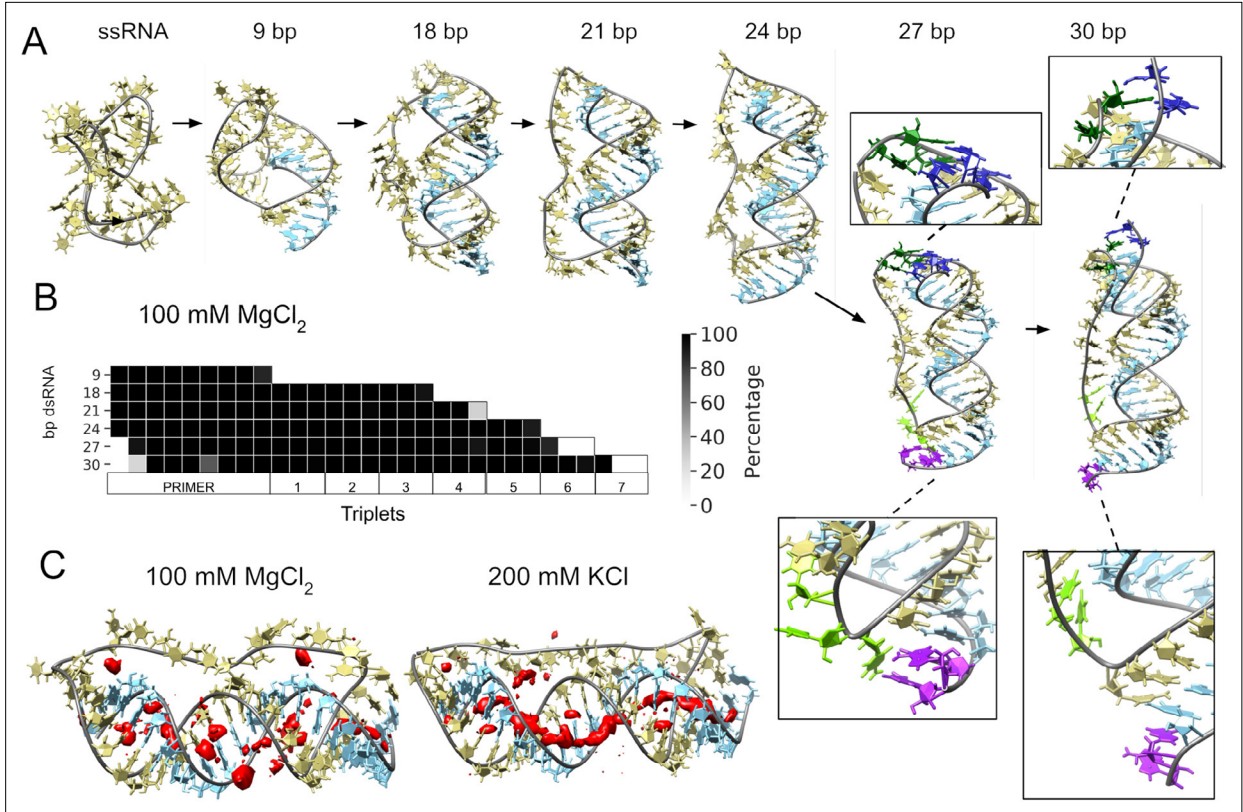

**Figure 3.** Molecular dynamics simulation of small circular RNA. (**A**) Main conformations (and zoom-in to relevant regions [squares]) observed from simulations in 100 mM MgCl$_2$ on scRNA exploring consecutive states of primer extension, from 9 to 30 bp dsRNA with pyrimidine (template) strand (UUC)$_{12}$ (khaki), purine (product) strand (GAA) (light blue), 5′-end and unpaired bases (dark blue) and 3′-end unpaired bases (purple) and matching melted bases from the template strand (dark green (5′-end)/light green (3′-end)). (**B**) Percentage of frames from the last 100 ns of the simulations presenting canonical hydrogen bond pairing for each bp. (**C**) Counterion-density maps (in red) around RNA molecules that show an occupancy ~10 times or greater than the bulk concentration.

The online version of this article includes the following figure supplement(s) for figure 3:

**Figure supplement 1.** Time sequence of snapshots from the simulation of scRNA with 27 bp dsRNA in 100 mM MgCl$_2$ with pyrimidine (template) strand (UUC)$_{12}$ (khaki), purine (product) strand (GAA) (light blue).

**Figure supplement 2.** Time sequence of snapshots from the simulation of scRNA with 30 bp dsRNA in 100 mM MgCl$_2$.

**Figure supplement 3.** Percentage of frames from the last 100 ns of the simulations presenting canonical hydrogen bond pairing for each base pair (bp).

**Figure supplement 4.** Roll, slide, twist, and major and minor groove widths.

**Figure supplement 5.** Counterion-density maps around RNA molecules that show an occupancy ~10 times or greater than the bulk concentration (in red) as seen in simulations.

**Figure supplement 6.** Averages and standard deviation of RDF of cations around RNA backbone phosphates.

introduction of kinks into the dsRNA segment (*Figure 3B*). Subsequently, multiple peeling off and rebinding events were observed during the trajectories indicating that the dynamics of this process are fast (*Videos 1 and 2* and *Figure 3—figure supplements 1 and 2*).

In the experimental data, we also observed an inhibitory effect for insertion of the final triplets (+8, +9, and +10 (beyond full length)/extension to 33, 36, and 39 nt of RNA in *Figure 2D*) into the corresponding scRNA template. This may indeed reflect the onset of the 3′- and 5′-ends destabilization observed in the MD simulations (*Figure 3*), which would likely attenuate primer extension by the ribozyme. Note however that the extension efficiency recovered beyond full length (+11/extension to 41 nt, *Figure 2F*), although at lower speed (*Figure 2E*).

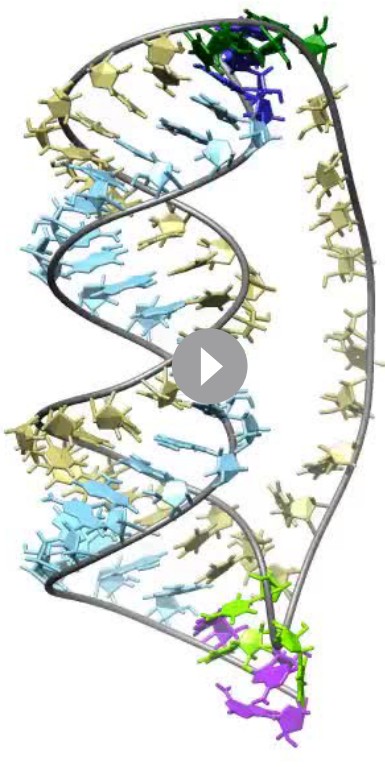 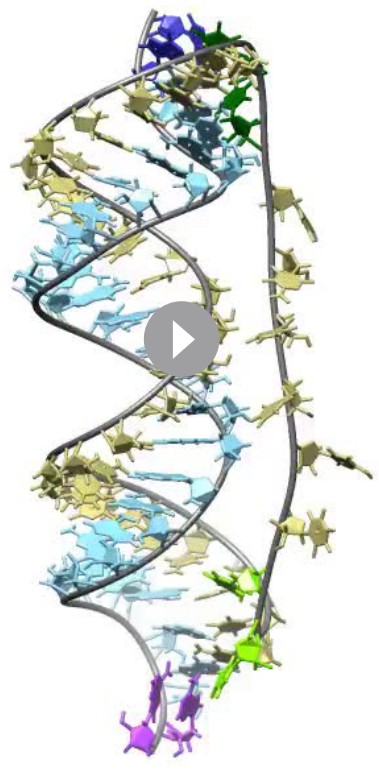

**Video 1.** Movie of the RCS simulation where dsRNA is 27-bp long. We observe fraying and annealing of 5'- and 3'-ends demonstrating the quick timescales of these transitions.

https://elifesciences.org/articles/75186/figures#video1

**Video 2.** Movie of the RCS simulation where dsRNA is 30-bp long. We observe again fraying and annealing of 5'- and 3'-ends demonstrating the quick timescales of these transitions.

https://elifesciences.org/articles/75186/figures#video2

As a control for the observed dsRNA end destabilization mechanism, we also ran an MD simulation of a linear RNA molecule containing four consecutive triplets with a nick between two of them, but observed neither base opening nor dissociation at either 5'- or 3'-strand ends (*Figure 3—figure supplement 3*). Groove dimensions and local helical parameters (roll, twist, and slide) for the RCS simulations on circular RNA did not show any major adjustment compared with the linear RNA control (*Figure 3—figure supplement 4*). We observed an oscillation of high/low values of bending along the molecule in phase with RNA-turn periodicity in an attempt to create an overall curvature (*Velasco-Berrelleza et al., 2020*), although with moderate success (~60° on an arc length of 30 bp of dsRNA) and no formation of kinks, internal loops or other disruption of the canonical A-form typical of the RNA duplex (*Figure 3—figure supplement 4*).

To mirror the experimental eutectic phase conditions, simulations were run at relatively high $Mg^{2+}$ concentrations (100 mM) and compared with the presence of monovalent ions like $K^+$ (200 mM) and high concentration of $Mg^{2+}$ (500 mM), but simulations did not show any major differences in terms of dsRNA strand dissociation or bending (*Figure 3—figure supplements 3 and 4*). However, $Mg^{2+}$ ions—compared to $K^+$—appear to interact more strongly with different parts of the RNA and, consequently, may increase the probability of distorted conformations facilitating the exposition of nucleobases at the 5'- and 3'-ends. In contrast, $K^+$ ions are mainly positioned as counterions along the major and minor groove, allowing the bases to orient toward the inside of the dsRNA helix for base-pairing interactions (*Figure 3C* and *Figure 3—figure supplement 5*). The role of $Mg^{2+}$ in the stabilization of complex RNA folding has been observed repeatedly in several structures (*Sponer et al., 2018*),

like the ribosome (*Klein et al., 2004*) and the Hepatitis delta virus ribozyme (*Nakano et al., 2001*). However, increasing MgCl$_2$ concentration to 500 mM does not seem to bring extra stabilization, as the system appears to be saturated already at 100 mM Mg$^{2+}$ (*Figure 3—figure supplements 5 and 6*).

In summary, our simulations support the notion that progressive RNA synthesis on a scRNA template (in the presence of Mg$^{2+}$ ions) leads to increased dynamics of nucleobase exposure, RNA duplex destabilization, and 5'- and 3'-ends melting. The simulations also clearly illustrate the implausibility of a small circular fully dsRNA molecule (as schematically illustrated in *Figure 1E*), due to the prohibitive energetic cost of dsRNA bending. Instead, the system appears to relieve internal strain by extending the ssRNA segment of the circle (partially shielding the dsRNA segment) and peeling of both dsRNA 5'- and 3'-ends (*Figure 3*), consistent with the helical period of triplet extension observed (*Figures 1 and 2*) (with ligation junctions facing into the ssRNA center being less accessible) and the observed reduction in RCS efficiency. Dynamic destabilization of dsRNA 5'-ends clearly has the potential to facilitate strand displacement during RNA replication on a scRNA template (and may aid continuous RCS), but at the same time may reduce the efficiency of primer extension and triplet incorporation by reducing the availability of the primer 3'-end and the downstream template bases. These effects would be predicted to manifest themselves in RNA circles up to 200 bp as suggested by RNA persistence length (*Abels et al., 2005*).

## Templated rolling circle RNA synthesis

Having validated RNA synthesis on scRNA templates (*Figures 1 and 2*), we next sought to show RCS beyond a single full-length circle synthesis involving displacement of the primer/nascent strand. To this end, we designed barcoded templates that would allow us to distinguish RNA synthesis products arising from template-instructed RCS from those arising from non-templated terminal transferase (TT) activity of the TPR by sequencing. The barcoded scRNA templates (termed A–D) were prepared either as circular or linear RNAs comprising different internal triplet 'barcodes' of variable GC-content (at positions 3, 6, and 9) for individual identification (*Figure 4A* and *Figure 4—figure supplement 1*). We performed one-pot primer extension experiments, in which all four templates (either A–D linear or A–D circular) were mixed in equal proportions. After extension, products were separated by gel electrophoresis and the gel sections above full-length extension products were excised and their RNA content recovered, and sequenced (*Figure 4—figure supplement 1*). Diagram in *Figure 4B* represents (in %) which triplets were identified at the noted positions. Boxes mark expected triplet according to the template sequence.

Analysis of the sequencing products from the one-pot reaction showed template-dependent high-fidelity RNA synthesis up to full length (position 9) for all templates (linear and circular) (*Figure 4B*). Further, all templates yielded longer than full-length products indicative of continued RNA synthesis by the TPR beyond full length (positions>9). However, the fidelity dropped after full length was reached, indicative of significant non-templated terminal transferase-like (TT) activity in this regime (*Figure 4B*). For example, the average fidelity for insertion of the expected triplet (^pppGAA) for position 10 (full length +1) for circular templates was 10.9% whereas for position 9 (full length) it was 89.9%. For linear templates, the fidelity for full length +1 dropped to 0.7% compared to full length 78.8%. Note, that fidelity at full length +1 dropped much more for linear than for circular templates. For this reason, the probability of a product extending to longer than full length (positions 10–12) with the correct sequence was 10–100-fold higher for circular compared to linear RNA templates (*Figure 4C*). A few events of blunt-end ligation with other template/product strands (see, e.g., position 15 for linear templates C and D) (*Figure 4B*) were also observed for linear templates.

On all circular templates (with the exception of template B, where too few reads were obtained) extension beyond full length (while containing a significant TT component) continued to insert the barcode triplets correctly, indicating continuous RCS at least up to position 18 (63 nt, more than 1.5 times full-length circle synthesis on the scRNA template). Control experiments, with individually incubated templates (in contrast to the one-pot experiments) mixed after gel purification, showed essentially identical results (*Figure 4—figure supplement 2A*). Interestingly, non-diluted samples had a decreased fidelity at position 10 (the point of strand invasion) compared to diluted samples (*Figure 4—figure supplement 2B*) suggesting that dilution appears to aid not only extension efficiency (*Figure 2—figure supplement 1D*), but also strand invasion fidelity and continued templated

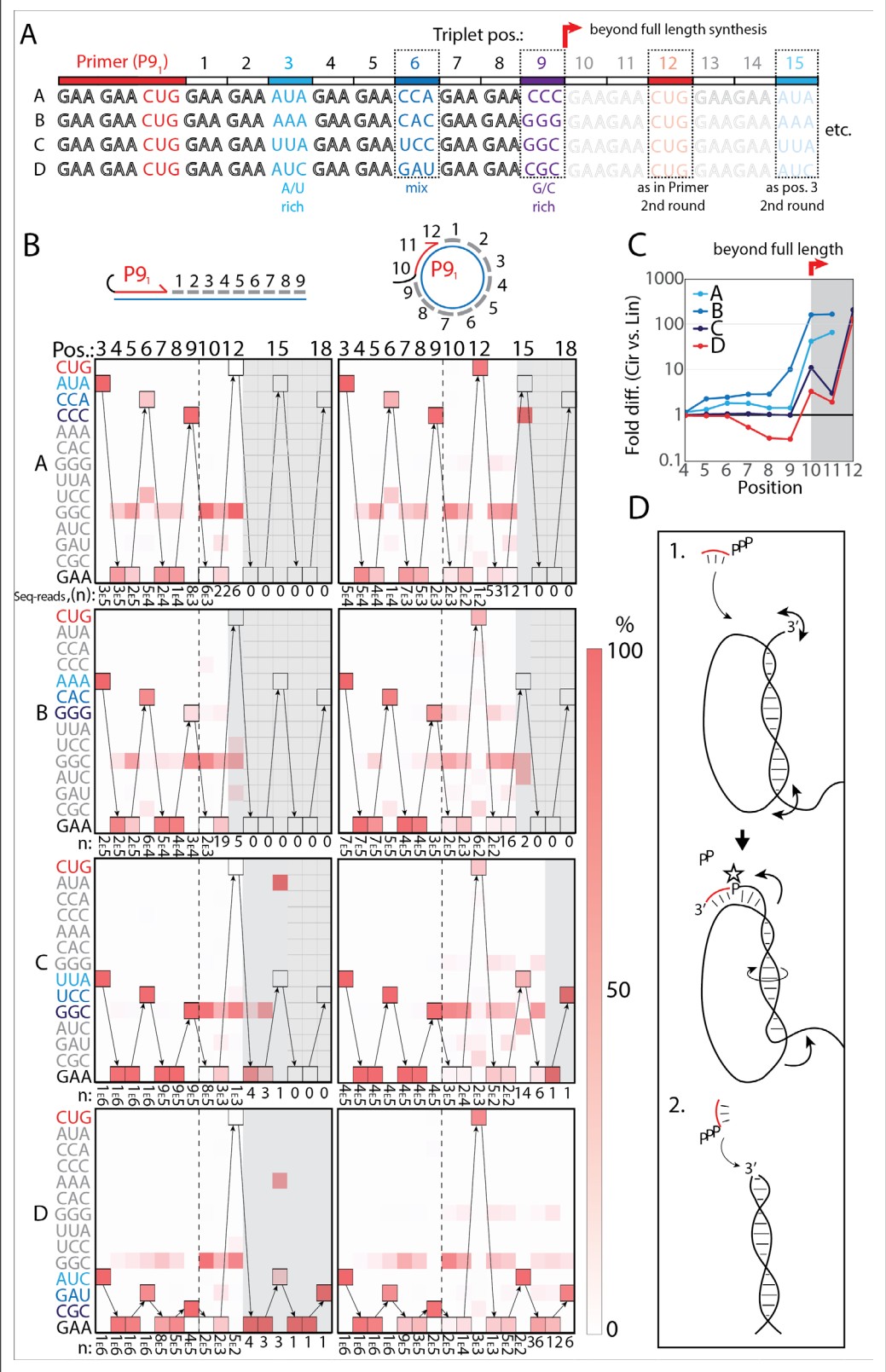

**Figure 4.** RNA-catalyzed RNA synthesis beyond full length for circular templates. (**A**) Product strands of primer extension experiments with linear and scRNA templates A–D with primer P9₁. Opaque sequence illustrate potential beyond full-length synthesis on scRNA. Barcode triplets at positions 3 (A/U rich) (cyan), 6 (mix) (blue), and 9 (G/C rich) (purple) allow identification of product RNAs. Barcode triplet at position 12 is the final triplet of primer

*Figure 4 continued on next page*

*Figure 4 continued*

$P9_1$ and at position 15 is the same as that of position 3 but after beyond full-length circle synthesis on the scRNA template. (**B**) Fidelity heatmap of the sequences derived from the one-pot experiments with linear (left) or circular (right) templates. Red color indicates high prevalence of a given triplet (vertical axis) at the position noted (3–18). n: denotes the number of sequence reads ($3_E5=3\times10^5$) used to calculate the fidelity for each triplet at the given position. Transparent gray boxes cover positions with n≤5. (**C**) Plot shows ratio (fold difference) of the probability of a product of reaching positions 4–12 on circular compared to linear templates. Fold difference was calculated based on fidelity data presented in (**B**). (**D**) Model illustrating (1.) beyond full-length extension on a circular template (templated RCS) and (2.) on a linear template (non-templated). Full analysis of the data in **B** is supplied in *Figure 4—source data 1*.

The online version of this article includes the following source data and figure supplement(s) for figure 4:

**Source data 1.** Full analysis of sequencing data used in *Figure 4B*.

**Figure supplement 1.** Deep sequencing of extension products.

**Figure supplement 1—source data 1.** Gel images.

**Figure supplement 2.** Controls for deep-sequencing data.

synthesis. In summary, these results are consistent with RNA-catalyzed RCS on scRNA templates beyond the full circle.

## Multiple repeat rolling circle products

Next, we sought to test if RCS efficiency could be increased by double priming on the circular RNA template, an approach known as branched RCS (*Berr and Schubert, 2006*). Indeed, we observed a higher degree of RCS with strand displacement with the 36 nt scRNA template termed sc8211 having two identical primer sites leading to two different products being formed (product I or II) (*Figure 5A and B*). In order to test the primer site functionality individually, we used different triplet combinations (*Figure 5B*). When only the ᴘᴘᴘGAA triplet was present, primers were extended only by two triplets as expected (lane 1 in *Figure 5B*) (with a small amount of non-templated TT incorporation of a third triplet). When ᴘᴘᴘGAA and ᴘᴘᴘAUA or ᴘᴘᴘGCG triplets were added, respectively (lane 2 or 3 in *Figure 5B*), products extended up to five triplet-incorporations with extension stopping at triplet 6 (coding for CUG) showing that both primer sites were functioning. Finally, when all triplets (ᴘᴘᴘGAA, ᴘᴘᴘAUA, ᴘᴘᴘCGC, and ᴘᴘᴘCUG) were present, extension continued to beyond full circle (positions≥10) (*Figure 5B*) and bands corresponding to extension products exceeding two times full-length circle synthesis (>triplet 21 (63 nt)) of replication were observed (*Figure 5C*).

Sequencing of the long, branched RCS RNA products (excised from gel band corresponding to ≥15 triplets, *Figure 5—figure supplement 1*) identified a range of long reads (from both products I and II) including many reads of the product with 15 correct triplet incorporations (*Figure 5E*) representing ~1.5 times full-length circle synthesis (n: $7\times10^3$ and $1\times10^5$ reads of products I and II, respectively). However, much longer sequences were present in decreasing numbers of reads, with the longest products comprising 29 correct triplet incorporations (96 nt) (n=2) representing RCS of >2.5 times full-length circle synthesis and the longest reported product synthesised by the TPR. Thus, RNA-catalyzed RCS has the potential to yield extended RNA concatemer products under isothermal conditions. Freeze-thaw (FT) cycles have been previously shown to enhance ribozyme activity by affecting RNA refolding (*Mutschler et al., 2015*) and indeed inclusion of four FT cycles lead to more efficient production of longer RCS RNA products (*Figure 5B and C*). In summary, isothermal branched RCS yields long concatemeric products comprising >2.5 tandem repeats of the scRNA template with RCS efficiency further enhanced by FT cycling.

## Proto-viroid like self-circularizing ribozyme

A number of biological systems including viroids use an RCS strategy for genome replication. However, RCS synthesis of RNA concatemers is only one part of the viroid replication cycle, which also involves resection (i.e., cleavage) of the concatemer into individual units and circularization of unit length RNAs by ligation to recreate the original circular RNA genome. As both RNA cleavage and RNA ligation can be efficiently catalyzed by RNA, we sought to investigate, if a viroid-like replication cycle might be catalyzed by RNA alone. To this end, we designed a proto-viroid RNA comprising a 39-nt

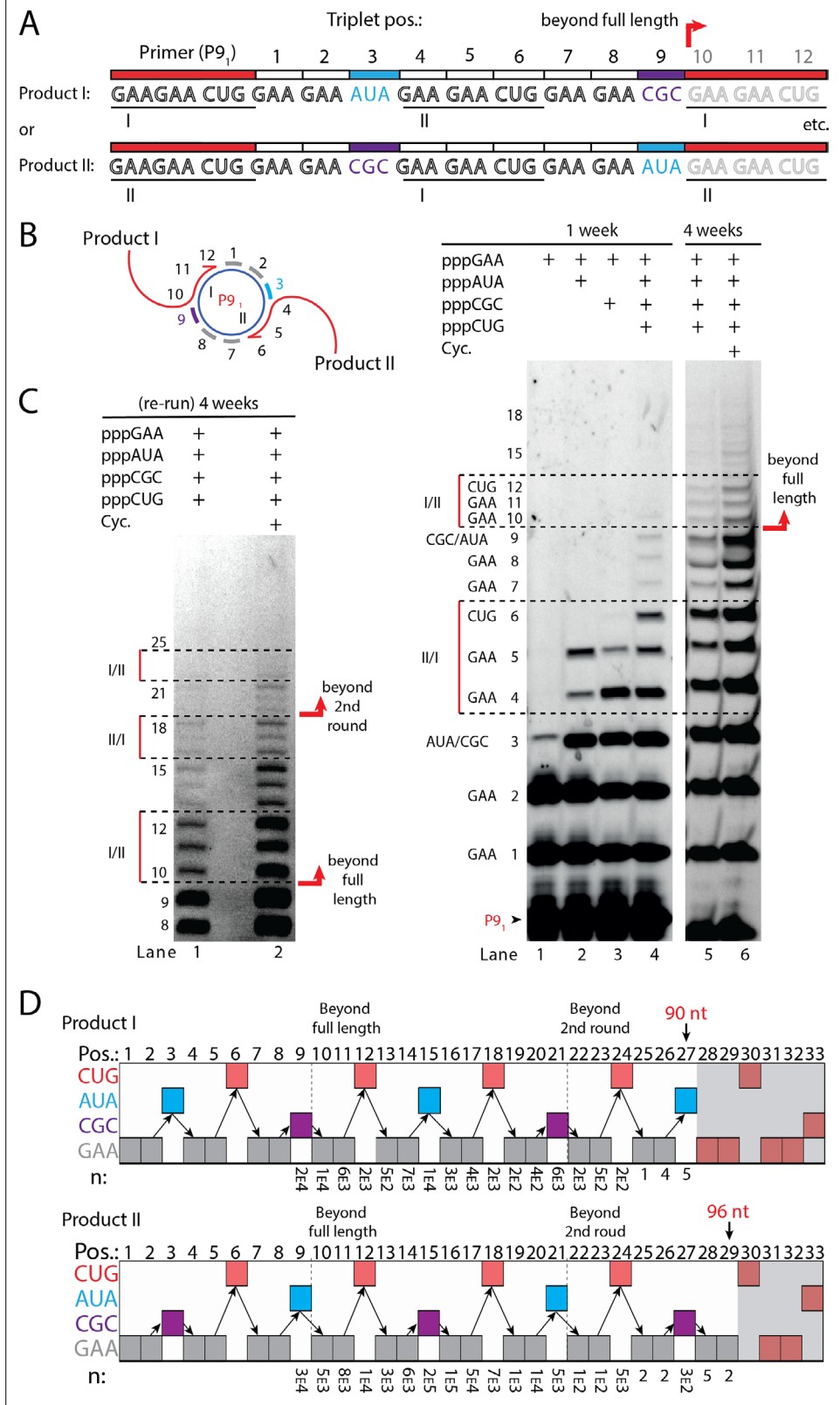

**Figure 5.** RNA-catalyzed branched RCS. (**A**) Product strand of primer extension experiments with scRNA template containing two priming sites (I and II) for primer P9₁. Depending on the priming site two different products will be made (I or II). (**B, C**) Scheme and PAGE of primer extension experiments with only the noted triplets added with (**C**) long electrophoretic separation to achieve optimal resolution of long products. Cycling (Cyc.) indicates that

*Figure 5 continued on next page*

*Figure 5 continued*

the samples had been exposed to four thermal and freeze-thaw cycles (80°C 2 min, 17°C 10 min, −70°C 5–15 min, and −7°C 7 days) leading to increased efficiency. (**D**) Sequencing of longer than full-length branched RCS products on the double primer site scRNA (without cycling). Products I and II both reaching almost three full rounds of replication of the circular RNA template (up to 96 nt, 32 triplet incorporations). Original gel images and full analysis of the data in **D** are supplied in *Figure 5—source data 1*.

The online version of this article includes the following source data and figure supplement(s) for figure 5:

**Source data 1.** Gel images and full data analysis of sequencing data used in *Figure 5D*.

**Figure supplement 1.** 10% Urea PAGE separation of circular template extension reaction used for deep sequencing.

---

scRNA template encoding a designed micro-hammerhead ribozyme (µHHz) as well as its substrate for cleavage (*Figure 6A*). We envisaged a viroid-like replication cycle (schematically illustrated in *Figure 6B*) comprising primer extension on a circular template (step 1) reaching full-length circle synthesis and beyond (RCA, step 2) resulting in formation of the µHHz and its 3′-end substrate. When separated from its template, the µHHz exists in (at least) two conformations (step 4) of which one is the active Hammerhead ribozyme cleaving the 3′-end substrate yielding a 2′,3′-cyclic phosphate (>p) (step 5). Finally, the ribozyme re-ligates (chemical mechanism illustrated in *Figure 6B*) to form a circular product strand (steps 6 and 7). The µHHz could be synthesized by the TPR (*Figure 6C*). Furthermore, the µHHz could catalyze both self-cleavage (forming a 2′,3′-cyclic phosphate (>p)) and re-ligation leading to circularization (under RCS reaction conditions at −7°C in eutectic ice) (*Figure 6D*, lane 2). A similar equilibrium between cleavage and ligation in eutectic ice had previously been observed for the unrelated hairpin ribozyme (*Mutschler et al., 2015*).

When the µHHz had been 5′-phosphorylated (*Figure 6D*, lane 3) only cleavage but no circularization was seen, as phosphorylation blocks the 5′-hydroxyl required as the nucleophile for re-ligation (see steps 5 and 6, *Figure 6B*). To the best of our knowledge, the µHHz is the smallest (39 nt) self-cleaving and -circularizing RNA system reported to date and the first-time self-circularization has been shown in a Hammerhead ribozyme. Kinetic analysis of the cleavage and circularization reaction show a slow but accumulating amount of cleavage product as a function of time (black points in *Figure 6—figure supplement 1A*). Analyzing the ratio between linear (cleaved) and circular (ligated) products (*Figure 6—figure supplement 1*) indicated that the proportion of circle was initially very high (approx. 40% after 12 hr). Based on this, it is likely that all or most µHHz molecules are transiently circular at some point immediately after cleavage, but become progressively trapped in a state unable to re-ligate, most likely due to hydrolysis of the >p or misfolding. While these steps only validate half of a full replication cycle (formation of a (+)-strand scRNA from a (−)-strand scRNA template), these results outline the potential for a full viroid-like rolling circle RNA replication cycle based on RNA-catalysis alone.

## Discussion

Viroids are transcriptional parasites composed entirely of a circular RNA genome and are considered the simplest infectious pathogens known in nature. They lack protein-coding regions in their genome and can be completely replicated in ribosome-free conditions (*Daròs et al., 1994*; *Diener, 2003*; *Fadda et al., 2003*; *Flores et al., 2009*). They can comprise a circular RNA genome of as little as ~300 nt in, for example, *Avsunviroidae* encoding a Hammerhead ribozyme responsible for maturation by resecting the replicated RNA genome (*Flores et al., 2000*). The resected viroid genome is then ligated (circularized) by a host protein (e.g., tRNA ligase) (*Nohales et al., 2012*). Due to the simplicity of this replication strategy, viroids have been suggested to represent possible 'relics' from a primordial RNA-based biology (*Diener, 1989*; *Flores et al., 2014*). Indeed, circular RNA genomes would present a number of potential advantages for prebiotic RNA replication, including increased stability by end protection (*Litke and Jaffrey, 2019*) and a reduced requirement for specific primer oligonucleotides to sustain replication (*Attwater et al., 2018*; *Szostak, 2012*). A circular RNA genome also resolves the end replication problem, that is, the potential loss of genetic information from incomplete replication in linear genomes. Circular RNA structures can self-assemble from RNA mononucleotides through wet-dry cycling (*Hassenkam et al., 2020*) providing potential initiation

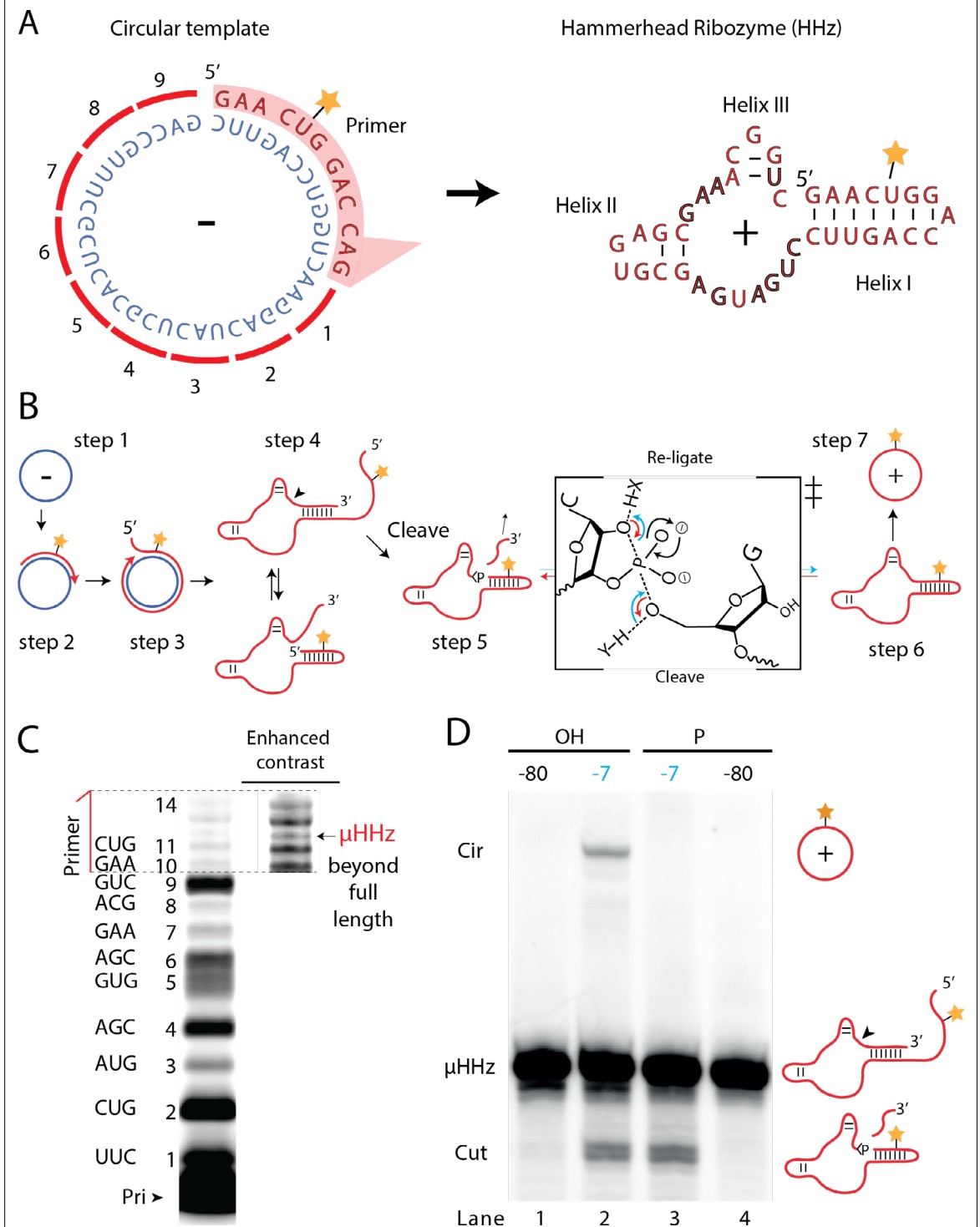

**Figure 6.** Steps of a viroid-like replication cycle catalyzed by RNA alone. (**A**) Illustration of the µHHz (−) and (+) strand. (**B**) Schematic illustration of the RNA-catalyzed viroid-like replication with steps comprising RNA-catalyzed combined RCS (steps 1–3), resection (steps 4 and 5), and self-circularization (steps 6 and 7). Panel between steps 5 and 6 illustrates the presumed chemical structures of the transition state of cleavage and re-ligation between the µHHz 3′- and 5′-ends. Blue arrow denotes the re-ligation reaction by bond formation with the 5′-nucleotide (forming the circle) and transesterification of the 2,3-cyclic phosphate bond (>p). Red arrows denote cleavage of the phosphate diester backbone (cleaving the circle) and formation of the 2′,3′-cyclic phosphate (>p). X and Y denote acids/bases involved in the transesterification reaction. (**C**) PAGE gel showing primer extension of RCS synthesis of the µHHz with substrate overhang to allow self-cleavage. scHHz temp was used as template and PHHz was used as primer (Pri). (**D**) PAGE

*Figure 6 continued on next page*

*Figure 6 continued*

gel showing cleavage and circularization of a µHHz, but only when incubated at –7°C, allowing eutectic phase to form, and with a free 5'-OH, needed for circularization, but not for cleavage. Original gel images are supplied in *Figure 6—source data 1*.

The online version of this article includes the following source data and figure supplement(s) for figure 6:

**Source data 1.** Gel images.

**Figure supplement 1.** Kinetic analysis of the µHHrz.

points for primordial RNA replication. Thus, viroid-like replication systems are plausible candidates to have emerged as the simplest genetic systems.

Here, we have explored to what extent such a potentially prebiotic replication strategy can be carried out by RNA alone. Our data show that RNA can indeed perform RCS on scRNA templates yielding concatemeric RNA products, which can be processed (i.e., resected) and re-circularized by an encoded ribozyme in a scheme reminiscent of viroid replication. Thus, one-half of a viroid replication cycle ((−)-strand replication leading to a self-circularizing (+)-strand) can be carried out by just two ribozymes. Completing a full viroid replication cycle would then require the reverse (+)-strand replication leading to a self-circularizing (−)-strand product. This may require a second ribozyme (e.g., a second µHHz) encoded in the (−)-strand akin to the mechanism used by natural viroids (*Flores et al., 2000*).

MD simulations indicate that the RCS process may be aided by accumulating strain in the nascent dsRNA segment leading to increased displacement (peeling off) of dsRNA 5'- and 3'-ends (i.e., strand displacement). In turn, this peeling off creates a more dynamic environment potentially aiding 5'-end invasion by extending the 3'-end. This topological strain-induced strand displacement may be general and independent of the precise RCS mechanism on scRNA templates and thus should also apply to non-enzymatic polymerization of RNA. Our observation that RCS can be enhanced by the use of branched extension, FT thermocycling, and pre-freezing dilution may also be related to this. Indeed, while the precise mechanistic basis for these enhancements is currently unknown, it seems plausible that all of these enhance 5'-end displacement on the product strand by accelerating conformational equilibration through RNA un- and refolding as observed previously (*Mutschler et al., 2015*).

In biology, both viroids and Hepatitis D virus (HDV) replication proceeds through RCS on circular RNA genomes mediated by proteinaceous RNA polymerases, but RCS has also been reported for circular DNA templates and proteinaceous DNA polymerases in nature (*Wawrzyniak et al., 2017*) and in biotechnology (*Daubendiek et al., 1995*; *Givskov et al., 2016*; *Kristoffersen, 2017*; *Mohsen and Kool, 2016*). dsDNA persistence length is somewhat shorter than dsRNA (dsDNA: 45–50 nm [140–50 nt] vs. dsRNA 60 nm [200 nt]) and stacking interactions in dsDNA are weaker than in dsRNA (*Kebbekus et al., 1995*; *Svozil et al., 2010*). Therefore, dsDNA may more readily adopt a circular shape or allow internal kinks to alleviate build-up of strain or to adopt strong bends (*Wolters and Wittig, 1989*). Nevertheless, we would expect a similar strand displacement effect would play a part in RCS on small circular DNA templates. Indeed, in both cases, RCS proceeds efficiently for circular genomes ranging from a few hundred nt to over 1.5 kb (*Mohsen and Kool, 2016*). In contrast, RNA-catalyzed polymerization (current maximum approx. 200 nt products; *Attwater et al., 2013*) is currently limited to RCS on scRNAs. A more efficient RNA-catalyzed RCS-based replication strategy will likely require improvements to the ribozyme polymerase catalytic activity, speed, and processivity as well as the design of the template. Improvements to ribozyme polymerase processivity, which is known to be poor (*Johnston et al., 2001*; *Lawrence and Bartel, 2003*), might have the greatest impact and might be realized either through tethering or other topological linkages to the circular template (*Cojocaru and Unrau, 2021*). A more processive polymerase ribozyme should also result in less non-templated triplet terminal transferase activity, which appears to be a consequence of slow RCS extension and is likely aggravated by dissociation of the 3'-end. Thus, more efficient RCS may also require the stabilization of the 3'-end triplet junction in the ribozyme active site in the same way as primer/nascent strand termini are stabilized within the active sites of proteinaceous RNA- and DNA polymerase (*Chim et al., 2018*; *Houlihan et al., 2020*). Finally, introduction of secondary structure motifs in the RNA nascent strand might drive increased 5'-end dissociation (e.g., through formation of stable hairpin structures) relieving strain at the 3'-end.

Larger circular RNA templates might provide advantages for the RCS as they are less strained. They would also provide increased access to the internal face of the circle and might be able to encode the whole ribozyme itself. On the other hand, reduced torsional strain on the dsRNA would be expected to reduce strand invasion and 'peeling off' of the product strand. All of these factors merit detailed investigation.

Our motivation for investigating RNA-catalyzed RCS was as a potential solution toward the so-called 'strand separation problem,' the inhibition of RNA replication by exceedingly stable RNA duplex products. This form of product inhibition has both a thermodynamic component, that is, the high amount of energy required to dissociate RNA duplexes, and a kinetic component as RNA replication must outpace duplex reannealing, which is rapid unless duplex concentrations are low or reactions take place in a highly viscous medium (*He et al., 2017*; *Tupper and Higgs, 2021*). In this context, we reasoned that RCS might provide favourable properties: synthesis and strand displacement on a circular template can in principle proceed essentially iso-energetically as base-pairing (H-bonding/stacking) interactions broken at the 5′-end during strand displacement are continuously compensated for by new base-pairing interactions formed at the nascent strand 3′-end. In turn, this enables an open-ended formation of template-coupled stochiometric excess of single-stranded RNA product strand to encode functions to further aid replication as we show here with resection and recircularization by an encoded ribozyme.

In the course of this work, we discovered another property of RNA synthesis on circular RNA templates that might contribute toward overcoming the strand separation problem. MD simulations indicate that—at least—on scRNA templates—the build-up of strain in the nascent dsRNA segment could aid strand displacement (*Figure 3*). However, the MD simulations also suggest that strain is non-directional destabilizing both nascent strand 5′- as well as 3′-ends equally. Thus, the potential advantages of scRNA RCS seem to be tempered by opposing effects such as strain as well as reduced template accessibility due to circular RNA ring geometry (*Figure 1*). Nevertheless, we find that a viroid-like replication strategy can be accomplished by RNA catalysis alone, with one ribozyme performing RCS on circular RNA templates yielding concatemeric RNA products, which can be processed (i.e., resected) and recircularized by a second ribozyme. Future improvements in polymerase ribozyme activity and processivity may allow all necessary components of such a replication cycle to be encoded on a circular RNA 'genome' and propagated by self-replication and -processing reactions.

## Materials and methods
### Oligonucleotides

Base sequences of all oligonucleotides used throughout this work can be found in *Supplementary file 1*.

In vitro transcription dsDNA templates (containing T7 promotor sequence at the 5′-end upstream of the region to transcribe) for in vitro transcription was generated by 'fill-in' using three cycles of mutual extension using GoTaq HotStart (Promega, Madison, WI) between the relevant oligonucleotide and primers: 5T7 or HDVrt (the latter for defined 3′ terminus formation *Schürer et al., 2002*). The T7 transcription protocol used is based on Megascript. Briefly explained, transcriptions of RNA requiring a triphosphate at the 5′-end (termed GTP Transcription) reaction were carried out under the following conditions: 40 mM Tris-Cl pH 8, 10 mM DTT, 2 mM spermidine, 20 mM MgCl2, 7.5 mM each NTP (Thermo Fisher Scientific), dsDNA templates (varying amount, preferably >5 µM), 0.01 units/µl of inorganic pyrophosphatase (Thermo Fisher Scientific, Waltham, MA), and ~50 µg/µl of T7 RNA polymerase (homemade by Isaac Gallego). Reactions were left overnight (~16 hr) at 37°C. Then 0.5×volume EDTA (0.5 M) was added together with (at least) 2.5×volume of loading buffer (final conditions >50% formamide or >8 M Urea and 5 mM EDTA). For transcription of RNA with a mono-phosphate 5′-end (termed GMP Transcription), the same procedure is followed as for NTP Transcription, however, 10 mM GMP and 2.5 mM of each NTP instead of the higher amount of NTP used for GTP transcription.

### Gel electroporation for analysis or purification

The sample in the appropriate loading buffer was separated on 10%–20% 8 M Urea denaturing PAGE gel using an EV400 DNA Sequencing Unit (Cambridge Electrophoresis). The product band

was visualized by UV shadowing (for non-labeled RNA) or fluorescence scanning (Typhoon scanner, Amersham Typhoon) (for labeled RNA). When needed the identified product based on relative migration was excised. The excised gel fragment was then thoroughly crushed using a pipette tip and suspended in 10 mM Tris-Cl pH 7.4 to form a slurry. For freeze and squeeze extraction, the slurry was frozen in dry ice, then heated to 50°C (~5 min), and finally left rotating at room temperature (from 2 hr to overnight) to elute the product from the gel material. The eluate was then filtered using a Spin-X column (0.22 µm pore size, Costar), ethanol precipitated, (100 mM Acidic acid and 80% ethanol (10 µg glycogen carrier was present when noted)). UV absorbance was measured with a Nanodrop ND-1000 spectrophotometer (Thermo Fisher Scientific) to determine yield of redissolved purified RNA.

## Calculation of extension efficiency

Gel images from the Typhoon scanner were analyzed in ImageQuant software (Cytiva Life Science) for quantifying band intensity. Quantified band intensities were exported to Excel (Microsoft, Redmond, WA) for further analysis. Extension efficiency (E) for a given band (b) was calculated as the sum of the intensities (I) of all the bands from b to n (n being the highest detectable band), divided by the sum of I of all bands from b−1 to n:

$$E_b = \frac{\sum_b^n I_b}{\sum_b^n I_{b-1}} \tag{1}$$

Thus, E represents the efficiency of the given ligation junction ($L_b$) to allow the production of the extension product in band b, that is, the extension efficiency.

## Triplet transcription

Triplets were prepared via run-off in vitro transcription with T7 RNA polymerase. More details on the method can be found in *Attwater et al., 2018*. Reaction conditions were as follows: 100 pmols of DNA template for each triplet was mixed with equimolar DNA oligo 5T7 to form the template for transcription. For triplets starting with purines, the NTP transcription protocol was used as described above with a total NTP concentration of 30 mM but only adding the nucleotides necessary for the triplet (e.g., AUA was transcribed with only ATP and UTP). For triplets starting in pyrimidines, a lower total NTP concentration was used (4.32 mM) as this yielded better defined bands for purification. About 50 µl transcription reactions were stopped with 2 µl EDTA (0.5 M) and 5 µl of 100% glycerol was added to facilitate gel loading. The samples were separated by 30% 3 M Urea denaturing PAGE as described above. Correct sequence composition was confirmed by A260/280 absorbance ratio, measured with the Nanodrop.

## Circularization of RNA

Linear 5'-end monophosphate labelled RNA to be used for circularization was either prepared by in vitro transcription (300 µl reaction volume) or ordered directly as chemically synthesized RNA (Integrated DNA Technologies [IDT], IA). Linear RNA was gel purified as described above. When needed purified RNA was treated with T4 polynucleotide kinase (PNK) (New England Biolabs (NEB), Ipswich, MA) to remove 3'-end cyclic phosphate then RNA was phenol/chloroform extracted, ethanol precipitated, and redissolved in ddH2O. For splinted ligation, 3 pmol purified RNA was mixed with equimolar splint RNA in 262.5 µl ddH2O. The sample was heated to 80°C (2 min) followed by cooling to 17°C (10 min) and finally incubated on ice (5–30 min). Then reaction conditions were adjusted to 50 mM Tris-HCl pH 7.5, 2 mM MgCl2, 1 mM DTT, and 400 µM ATP (1× T4 RNA ligase 2 reaction buffer [NEB]) including 0.25 units/µl T4 RNA ligase 2 (Neb) (final volume 300 µl) and samples left overnight (~16 hr) at 4°C. For non-splinted ligation, 0 pmol gel purified RNA was mixed in 237 µl ddH2O followed by heating to 95°C and then quickly moved to ice. Then reaction conditions were adjusted to 50 mM Tris-HCl, pH 7.5, 10 mM MgCl2, 1 mM DTT (1× T4 RNA ligase reaction buffer [Neb]), 100 µM ATP including one unit/µl T4 RNA ligase 1 (NEB) (final volume 300 µl) and samples left overnight (~16 hr) at 16°C. Circularized RNA was electrophorated by 10% 8 M Urea denaturing PAGE for analysis and purification as described above.

## Templated RNA-catalyzed RNA synthesis (the primer extension assay)

Ribozyme activity assay was performed essentially as described in *Attwater et al., 2018*. In a standard reaction (modified where specified), ribozyme heterodimer (5 TU/t1), template, primer (5 pmol

of each), and triplets (50 pmol of each) were annealed in 7.5 µl water (80°C 2 min, 17°C 10 min). Then 2.5 µl 4× reaction buffer was added (final volume 10 µl) and samples were left on ice for ~5 min to ensure folding. Final pre-frozen conditions were (unless otherwise noted) either (Tris buffer system) 50 mM Tris (pH 8.3 at 25°C), 100 mM MgCl2, and 0.01% Tween 20, or (CHES buffer system) 50 mM CHES (pH 9 at 25°C), 150 mM KCl, 10 mM MgCl2, and 0.01% Tween 20. At this point some samples (noted in the text) were diluted by adding ddH2O (e.g., 50× dilution corresponds to adding 490 µl ddH2O to the 10 µl samples). Finally, samples were frozen on dry ice and incubated at –7°C in an R4 series TC120 refrigerated cooling bath (Grant, Shepreth, UK) to allow eutectic phase formation and reaction, respectively. To end the incubations, samples that had been diluted were thawed, moved to 2 ml tubes, ethanol precipitated (with glycogen carrier) and redissolved in 10 µl ddH2O. This step was avoided for undiluted samples that were already 10 µl. Finally, 0.5 µl EDTA (0.5 M) was added to all samples to a final volume of 15 µl. (In experiments where the effect of dilution was investigated, e.g., as experiment presented in Supporting *Figure 5*, ddH2O was added to all the thawed samples to reach the same volume before precipitation). To prepare for separation of extension products, 3 µl of the reacted samples after the addition of EDTA (corresponding to 1 pmol template RNA) was diluted to reach the final loading conditions: 166 mM EDTA, 6 M Urea (+Bromophenol blue), and 10–20 pmol competing RNA (to prevent long product/template reannealing) (final volume 10 µl). Finally, samples were denatured (95°C for 5 min) and RNA separated by 8 M Urea denaturing PAGE.

## Sequencing of extension products

In the primer extension reactions used for sequencing, the primer extension was performed as described above except for the following changes: 5 pmol ribozyme heterodimer/template, 20 pmol primer (with a 5′ adapter sequence), and 100 pmol of each triplet was used. In the cases where multiple templates were mixed in the same reaction (one-pot), the final template concentration remained 5 pmol in total. All reactions were done in the CHES buffer and were diluted 50× as standard.

### Adapter ligation and RT-PCR

After Urea PAGE separation of the extension products, the noted region of the gel was dissected out, and carefully recovered as described above. The RNA was ethanol precipitated (80% ethanol with 10 µg glycogen carrier) resulting in a dry RNA pellet. To append an adaptor sequence to the 3′-end of the purified RNA products the dry RNA was redissolved in conditions allowing adenylated adapter ligation by T4 RNA ligase 2 truncated K227Q (Neb) following manufacturers' descriptions. Final adapter ligation conditions were: 50 mM Tris-HCl, pH 7.5, 10 mM MgCl2, and 1 mM DTT (1× T4 RNA ligase reaction buffer [NEB]), 15% PEG8000, 0.04% Tween 20, 5 pmol adenylated DNA primer (Adap1, for base sequences see *Supplementary file 1*), and 20 U/µl T4 RNA ligase 2 truncated K227Q (Neb) (final volume 10 µl). The samples were then ligated at 16°C for 2 hr. Pre-adenylation of Adap1 using Mth RNA Ligase (Neb) was performed following manufacturers' descriptions. After adapter ligation, samples were diluted tenfold to achieve conditions for performing RT-PCR (25 cycles) using 0.5 µM forward (PCRp3) and reverse primer (RTp1) and the SuperScript III One-Step RT-PCR System with Platinum Taq DNA polymerase (Thermo Fisher Scientific). Finally, RT-PCR products were gel purified in 3% agarose gel and cleaned up using QIAGEN gel extraction kit (QIAGEN, Hilden, Germany).

### Sanger sequencing

Purified RT-PCR products were cloned into pGEM vector using pGEM-T Easy Vector Systems (Promega) as described by the manufacturer and transformed into heat-competent 10-Beta cells (NEB). Inserts from single colonies were PCR amplified (using primers pGEM_T7_Fo and pGEM_SP6_Ba) and send in for Sanger sequencing (Source Bioscience) (using pGEM_T7_Fo as sequencing primer).

### Illumina sequencing

Illumina adaptors were added to purified RT-PCR products by PCR (15 cycles) using 0.5 µM forward (Illx_Fo, x denotes different barcodes 1–15, see oligo sequences in *Supplementary file 1*) and reverse primers (Ill_Ba) and Q5 Hot-Start High-Fidelity 2X Master Mix (Neb). PCR products were gel purified in 3% agarose gel and qPCRed (using NEBNext Library Quant Kit for Illumina) to quantify concentration.

Finally, the DNA (consisting of Illumina adapters, barcodes, and RT-PCRed sequence from the RNA extension) were prepared following manufactures protocol for MiSeq Illumina sequencing (Illumina, San Diego, CA) (see, e.g., MiSeq System Guide).

## Sequencing data analysis

Illumina Sequencing data were acquired and processed as FASTQ files using Terminal (and available software packages such as FASTX-toolkit). Prior to analysis, the whole output file from Illumina sequencing (containing also unrelated sequences) was split based on barcodes identifying the individual samples and trimmed starting with the original ($P9_1$) primer sequence (GAAGAACTG). After the $P9_1$ sequence, the triplets at positions 1, 2, 3, and so on, would be identifiable representing extension products made by the ribozyme. The presence of the 3′ adapter sequence (GTCGAATAT…) in the aligned sequences marked the end of the original RNA extension product. Sequencing data can be found as described below under section data availability: File 1 includes sequence data for circular and linear one-pot analysis (C1 and L1, respectively), File 2 includes sequence data for branched RCS analysis (B3).

### Analysis of the one-pot experiments

By counting the number of times a given triplet was present at a given position, we were able to calculate the fidelity for each triplet at this position. Identifying and counting the sequencing reads (n) for each position was done using *grep* (in Terminal) with a list of all relevant sequences (positions 3–18) and the sequencing files. The triplet at position 3, the first barcode position, was used to classify the sequences into coming from templates A to D and thus has 100% fidelity for the correct triplet (*Figure 4B*).

For example, for analyzing the fidelity (F) of position 4, the following list was used: GAAGAACT-$G_{(primer)}$GAA$_{(pos1)}$GAA$_{(pos2)}$YYY$_{(pos3)}$XXX$_{(pos4)}$. Here, YYY was either of the first barcode triplets for templates A–D, (ATA$_{(template A)}$, AAA$_{(template B)}$, TTA$_{(template C)}$, or ATC$_{(template D)}$) and XXX was either of the 14 possible triplets (CTG, ATA, CCA, CCC, AAA, CAC, GGG, TTA, TCC, GGC, ATC, GAT, CGC, and GAA). F at position 4 was then calculated for templates A–D as the number of occurrences of a triplet in position 4 (e.g., CCA) divided by the sum of occurrences of all the triplets multiplied by 100%. A generalized term for calculating the F at all positions (3–18) and for all templates (A–D) is:

$$F_{(a,Y)} = \frac{n_{(xxx,a,Y)}}{\sum \left( n_{(XXX,a,Y)} \right)} \times 100 \qquad (2)$$

Here, F is the fidelity, a is the position of the triplet, Y is the templates A–D. n is the number of sequencing reads for a given triplet (xxx) for position a on template Y or for all the 14 triplets (XXX), for a on Y. Eventually, the fidelity for positions 3–15 in the context of templates A–D for all triplets was plotted in *Figure 4B*. Accumulated chance for a product of reaching position X (shown in plot in *Figure 4C*) was calculated by multiplying all fidelities for moving from position three to position X with correct triplets (fidelities found in *Figure 4C*). Data for this analysis can be found as described in the Data availability section below (File 1). Numerical data and calculation are supplied in *Figure 4—source data 1*.

### Analysis of the branched RCS

By counting the number (n) of correct sequences with a specific length ending in the 3′ adapter sequence, we identified long RCS products (*Figure 5D*). This was done using *grep* (in Terminal) with a list of all relevant sequences (positions 9–30, both products I and II), and the sequencing file. Data for this analysis can be found as described in the Data availability section below (File 2). Numerical data and calculation are supplied in *Figure 5—source data 1*.

## Self-circularizing micro hammerhead ribozyme assay

RNA-catalyzed synthesis of fluorophore labeled self-circularizing µHHz was prepared in 2× large (500 pmol) reactions set up and incubated as described above. Specifically, 500 pmol ribozyme heterodimer (5 TU/t1) and circular template (scHHz_temp), 2000 pmol primer (HHrzP12) and 50 µmol of each of the triplets were annealed followed by adding buffer to 50 mM CHES, pH 9, 150 mM KCl, 10 mM MgCl2, and 0.05% Tween 20 (1 ml). Then the sample was diluted 50 times to a final volume of

50 ml. After 4 weeks incubation at –7°C, EDTA was added (5 mM final concentration), reactions were thawed and concentrated to a final volume of ~300 µl using a centrifugation filter (Amicon Ultra, 3 kDa cutoff) retaining long RNA products. µHHz RNA (marked in *Figure 6C*) was purified by gel electrophoresis and excised product was dissolved to 10 µM in H2O with 0.5 mM EDTA.

Chemically synthesized fluorophore labeled self-circularizing µHHz RNA (IDT) was gel purified as described above and excised product was dissolved to 10 µM in H2O with 0.5 mM EDTA.

## Micro-hammerhead ribozyme cleavage/circularization assay

Self-circularization assays of chemically synthesized fluorophore labeled µHHz comprise 10 pmol *µHHz* annealed (80°C 2 min, 17°C 10 min) in 4 µl water with 1 µl 5× reaction buffer, final reaction conditions: 50 mM CHES, pH 9, 150 mM KCl, and 10 mM MgCl2 (same as for the templated RNA-catalyzed RNA synthesis). Then incubated in ice for 5 min to ensure folding. This was then frozen on dry ice and either moved to –7°C for eutectic phase formation (reaction) or –80°C (control). After incubation, 10 µl loading buffer (95% formamide, 25 mM EDTA, and bromophenol blue) was added directly to the cold samples to stop the reaction and mixed while thawing. Finally, reactions were analyzed by 20% denaturing PAGE like described above. 5′-phosphorylation of µHHz RNA with polynucleotide kinase (NEB) was carried out following the manufacturer's directions. RNA was then phenol/chloroform extracted, precipitated, and dissolved in ddH2O with 0.5 mM EDTA to 10 µM (determined by Nanodrop).

## Molecular dynamics simulations

All simulations were set up with the AMBER 18 suite of programs and performed using the CUDA implementation of AMBER's pmemd program (*Case, 2018*). A linear ssRNA of 36 nt with the sequence $(UUC)_{12}$ was built using the NAB utility, which was then circularised using an in-house program (*Pyne et al., 2021*). From there, the complementary strand containing GAA triplets was progressively grown representing the different stages of the rolling circle replication, containing 9, 18, 21, 24, 27 till 30 nt of dsRNA keeping the rest single-stranded. For each stage, a representative structure was used as a scaffold to grow the dsRNA part and thus build the structure to model next stage. A linear dsRNA fragment containing four GAA triplets with a nick between the first and second was run as a control. This molecule had a total length of 16 bp as it was capped by a CG dimer on each end.

The AMBER99 forcefield (*Cheatham et al., 1999*) with different corrections for backbone dihedral angles including the parmBSC0 for α and γ (*Pérez et al., 2007*) and the parmOL3 for $\chi$ (glycosidic bond) (*Zgarbová et al., 2011*) were used to describe the RNA. All initial structures were explicitly solvated using a truncated octahedral TIP3P box with a 14-Å buffer. They were neutralized by two different types of salt, KCl, and MgCl2, described by the 'scaled charged' Empirical Continuum Correction (ECC) set of ion parameters (*Duboué-Dijon et al., 2018*), which substantially reduce the overestimation of ion-ion interactions with respect to water-mediated interactions typical of empirical forcefield calculations (*Fingerhut et al., 2021*; *Kirby and Jungwirth, 2019*) (see *Figure 3—figure supplement 6*). The necessary ion pairs (*Machado and Pantano, 2020*) were added for matching 0.2 M in the case of KCl, and 0.1 and 0.5 M in the case of MgCl2. Simulations were performed at constant T and P (300K and 1 atm) following standard protocols (*Noy and Golestanian, 2010*) for 400 ns.

The last 100 ns sampled every 10 ps were used for the subsequent analysis. AMBER program CPPTRAJ (*Roe and Cheatham, 2013*) was used to determine base-pair step parameters, radial distribution functions of ions around RNA and distances between atoms, including groove width and hydrogen bonds. The latter were defined with a distance cutoff of 3.5 Å and an angle cutoff of 120°. Counterion-density maps were obtained using Canion (*Lavery et al., 2014*) and were subsequently visualized with Chimera (*Pettersen et al., 2004*). SerraNA software was used to calculate curvatures at different sub-fragment lengths (*Velasco-Berrelleza et al., 2020*).

## Acknowledgements

This work was supported by the Carlsberg Foundation (CF17-0809) (ELK), by the Medical Research Council (MRC) program Grant program no. MC_U105178804 (PH), by the Engineering and Physical Sciences Research Council (EPSRC) Grant EP/N027639/1 (AN) and by the EPSRC (EP/R513386/1) (MB). Simulations were performed on JADE (EP/T022205/1). The authors thank the HecBiosim consortium

(EP/R029407/1), Cambridge Tier-2 (EP/P020259/1), and the local York facilities. Correspondence and requests for materials should be addressed to PH.

## Additional information

### Funding

| Funder | Grant reference number | Author |
| --- | --- | --- |
| Carlsbergfondet | CF17-0809 & CF19-0019 | Emil Laust Kristoffersen |
| Medical Research Council | MC_U105178804 | Philipp Holliger |
| Engineering and Physical Sciences Research Council | EP/N027639/1 | Agnes Noy |
| Engineering and Physical Sciences Research Council | EP/R513386/1 | Matthew Burman |
| Engineering and Physical Sciences Research Council | EP/T022205/1 | Agnes Noy |

The funders had no role in study design, data collection and interpretation, or the decision to submit the work for publication.

### Author contributions
Emil Laust Kristoffersen, Writing – original draft, Writing – review and editing; Matthew Burman, analysed data, discussed results and co-wrote the manuscript; Agnes Noy, analysed data, discussed results and co-wrote the manuscript, performed dynamics simulation, analysed data, discussed results and co-wrote the manuscript; Philipp Holliger, analysed data, discussed results and co-wrote the manuscript, analysed data, discussed results and co-wrote the manuscript

### Author ORCIDs
Emil Laust Kristoffersen http://orcid.org/0000-0001-8965-8201
Agnes Noy http://orcid.org/0000-0003-0673-8949
Philipp Holliger http://orcid.org/0000-0002-3440-9854

### Decision letter and Author response
Decision letter https://doi.org/10.7554/eLife.75186.sa1
Author response https://doi.org/10.7554/eLife.75186.sa2

## Additional files

### Supplementary files
- Transparent reporting form
- Supplementary file 1. Oligonucleotide sequences.

### Data availability
All data generated or analyzed in this manuscript is supplied within the manuscript or supporting file; Source Data files containing original unedited gels images as well as numeric data have been provided for Figures 1,2,4 and 5, as well as figure supplements when relevant. Modelling data and sequencing data are provided as described in the data availability section in the manuscript.

The following dataset was generated:

| Author(s) | Year | Dataset title | Dataset URL | Database and Identifier |
| --- | --- | --- | --- | --- |
| Kristoffersen EL | 2021 | Deep Sequencing data for document titled: Rolling Circle RNA Synthesis Catalysed by RNA | https://doi.org/10.5061/dryad.tht76hf10 | Dryad Digital Repository, 10.5061/dryad.tht76hf10 |

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
