## [Editor Report]

This paper is of interest to scientists from the field of origin of life or RNA synthesis in general, especially those interested in the "RNA world" scenario. The data analysis is rigorous and the conclusions are justified by the data. The key claims of the manuscript are directly related to, and support, previous findings.

---

## [Decision Letter]

**Decision letter after peer review:**

Thank you for submitting your article "Rolling Circle RNA Synthesis Catalysed by RNA" for consideration by *eLife*. Your article has been reviewed by 3 peer reviewers, and the evaluation has been overseen by Timothy Nilsen as Reviewing Editor and James Manley as the Senior Editor. The following individual involved in review of your submission has agreed to reveal their identity: Jiri Sponer (Reviewer #3).

Essential revisions:

*Reviewer #1:*

In the origin of life scenario where RNA is assumed to be the first replicator, a key problem is how RNA can replicate itself. Or how can RNA polymerase copy itself, since copying requires an open flexible structure that can be read. While the polymerase needs to be a topological rigid structure in order to catalyse the RNA polymer.

The manuscript describes how a special kind of synthesis of the RNA, rolling circle synthesis on small pieces of circular RNA, could template and build RNA strands. The method apparently could help in avoiding the strand inhibition problem where stable RNA duplex in long strands hinders replication.

It is a bit unclear how, but I suspect this i down to the size of the ring, since the problem increase with the length of the polymer. The authors also touch on this themselves in their MD simulations, since the entropically unfavourable confinement of a string onto a circle can alleviate part of the problem. Nevertheless, the authors also show by MD that the rings become small tight structures that actually hinders replication.

The study depends on trinucleotide triphosphates (triplets) as substrates. They demonstrate a viroid like replication that show how a template (-) is able to make a mirror copy (+) of circular RNA by a rolling circle synthesis with out the need for enzymes or other catalyst apart from RNA itself.

While the conclusion of the work becomes a bit muddled, but honest, the work is a very important piece that demonstrates the huge potential role of circular RNA in the very early stages of life.

I must confess I am very far off my field of competences. I have tried my best to understand the paper, and the methods involved, but obviously I cannot give much feedback on the methods used. So, I cannot suggest improvements on that part.

For the most part, the paper is well written, and the structure is sound. I would strongly recommend publication.

It was unclear to me how it was certain that it was actually rings. There is the MD simulations, but apart from the principle drawing they are obviously not circular in shape.

There are some parts of the manuscript that too me is very difficult to understand. Some statements seem to reflect a community understanding that might be obvious for those working with similar system on a day to day basis, but for the general reader (where I put myself) this becomes gibberish. for instance:

RCS has potentially unique properties with regards to the strand inhibition problem where RNA duplex melting in principle can be effected by continuous toehold strand displacement driven by nucleotide hybridization and the ratchet of nascent strand extension by triphosphate hydrolysis. In an idealized RCS mechanism, such strand invasion and displacement processes are both isoenergetic and coordinated to nascent strand extension (Blanco et al., 1989; Daubendiek et al., 1995), with rotation of the single-stranded RNA (ssRNA) preventing the build-up of topological tension (Kuhn et al., 2002). Thus, RCS is a potentially open-ended process leading to the synthesis of single-stranded multiple repeat products (concatemers) with an internally energized strand displacement circumventing the "strand inhibition problem" (Tupper and Higgs, 2021).

It might not be possible to rephrase this so that everyone can understand. But I think the clarity of manuscript could be improved.

But also because there is references to previous work where long statements does not make it more clear what is actually meant. for instance:

Similar to what was described previously, RNA synthesis by the TPR best in the eutectic phase of water ice, due to beneficial reaction conditions for ribozyme catalysis such as reduced RNA hydrolysis and high ionic and RNA substrate concentrations (Attwater et al., 2010). This was also the case on scRNA templates.

That said, I think the overall message is clear and the paper is very interesting, I expect to reference it when it comes out.

*Reviewer #2:*

The RNA World theory is one of the most widely-believed explanations for the origin of life. This relies on the idea that there were self-replicating RNA systems in the early stages of life. Usually ,it is supposed that there were polymerase ribozymes that were able to use another RNA strand as a template for synthesis of the complementary strand. As there are no naturally-occurring polymerase ribozymes, there has been a sustained effort over several decades to develop polymerase ribozmes in the lab by in vitro selection. This paper contributes to this by presenting a polymerase ribozyme that can copy a circular template. Circular templates are thought to be important because replication of a circular template can occur via the rolling circle mechanism, in which a polymerase continues multiple times around the same circle, and the far end of the growing strand is displaced from the template at the same time as new bases are added to the growing end. This avoids the problem of strand inhibition (i.e the difficulty of separation of stable double strands that are expected to form when copying linear templates).

This paper considers rolling circle replication on very short circles of around 36 nucleotides. It is shown that replication proceeds by addition of triplets beyond the full length of the circle. As the circle is short, and the double-stranded part is stiff, it is not possible for the whole of the circular template to be double-stranded at the same time. It is shown that roughly half of the circle is double-stranded, and that the separation of the two strands occurs at a point which is on the opposite side of the circle from the point of primer extension.

The rolling circle mechanism involves cleavage of the growing strand by a self-cleaving hammerhead ribozyme that is encoded in its sequence. The mechanism also requires the reconnection of the ends of the new strand in order to form a new circular template. Both the cleavage and re-circularization steps are demonstrated in this paper.

This experiment still falls short of a fully self-replicating ribozyme system, because in order for continued replication to occur, both plus and minus strands of the circle would have to encode a hammerhead ribozyme, and in order for the system to be self-sustaining, the circles would also have to encode the polymerase ribozyme itself (which is supplied separately in this paper and is not replicated). Nevertheless, this paper makes an important step, and continues to bring us closer to developing self-replicating RNA systems.

Lines 122-126 – It is implied that triplets are better than monomers for rolling circle replication because triplets help to open up other double stranded regions. However, it is not obvious that this should be the case. To put a new triplet down you have to displace three bases from the displaced strand, whereas to put a monomer down you only have to displace one base. It is not easy to predict which of these is faster without measuring it. Furthermore, in the actual mechanism occurring here, there is no prior strand to be displaced at the point of attachment, because the displacement is occurring at the other side of the circle and it does not directly interfere with the attachment. So it is not clear whether this argument applies. Has replication of a circular strand actually been attempted with a monomer ribozyme? Is it known whether a triplet ribozyme is better than a monomer ribozyme on circular templates? If not, it would be better to avoid implying this.

Figure 1 – the periodic effect seen in 1E is claimed to be due to the difference of accessibility of template bases on the inside and outside of the circle. However, the results are measured by averaging over many different circular templates. I would expect that different copies of a circular template would have different configurations and would not always have the same bases on the inside. So the inside-outside difference should average out. Could the variability of 1E be explained by variation in rates of addition according to the sequence of the template rather than an inside-outside effect? The sequence effect would be the same in multiple copies of the same template sequence. Is there a similar variability seen when copying linear templates?

Figure 1C shows a TPR dimer. Is the polymerase actually in two parts? Is this important?

Line 213 – It is not clear why the 9 bp primer goes straight to 18 bp. What happened to the lengths in between?

Line 220 – The word "extended" is used to mean that the unhybridized portion is stretched. There is a possible confusion with extending a strand by ligation of a triplet. Maybe the use of a word like stretched is better?

Line 226 – The simulations show that the double-stranded part of the circle is stiff and only covers roughly half of the circle. The point at which the primer extension occurs is therefore far away from the point at which the two strands separate. This is important for very short circles. For longer circles, the stiffness should be less relevant, and there will come a point where the whole circle becomes double stranded. There will then need to be a true strand displacement occurring very close to the point of primer extension. How long would we need the circle to be before it switches to a double-stranded circle? Does the stiffness effect seen here with the short circles make the primer extension reaction easier or more difficult than a true strand displacement reaction on a double stranded circle?

Lines 228-33 – This paragraph is not very clear. The meaning is not coming through.

Line 288 – "orbit" is an odd word. Is there a better one?

Figure 4 – Overall Figure 4 is not clear.

– I have not understood the notation n: 3E5, 2E5 etc.

– For sequence A Pos 4, GAA is the darkest shade, so I am presuming the template is CUU (in the reverse direction). But the second darkest shade is GGC. Why should GGC bind to CUU more strongly than others (for example GGA)? I am not sure whether I have understood this diagram correctly.

– Part C shows fold difference. It would be easier if rates where shown for linear and circular strands separately. Why is sequence D a worse template? Or maybe it is not worse – it's just that the ratio of circular to linear templates is lower? It is not easy to understand this.

– Part D Figure 2 seems to show a double-stranded triplet being added. Why not just a single-stranded triplet.

Figure 6D – It is unclear what is happening at each step. Particularly the backwards and forwards diagrams in step 4. Also, shouldn't the red strand be still attached to the blue circle before the cleavage occurs? The chemical structure in the middle is a bit distracting. I think the structure drawn in A is the same as D step 6. Maybe put parts A and D together and make B and C a separate figure?

Line 436 – The reference to the virtual circular genome is misleading at this point. In the proposal of Zhou et al., there are no real circles, there are simply linear fragments that can be aligned to form a virtual circle. This does not fit with the rest of this paragraph. Either the reference to Zhou et al. should be omitted or it should be explained properly what the virtual circle proposal is.

*Reviewer #3:*

Technically the experiments are sound, really comprehensive, convincing and the paper is well written. The documentation (the composed Figures etc.) is very nice. The MD simulations nicely complement the experiments. Strong point is that the simulations address a qualitative question and are clearly directed to solve it. It is a preferable application of the MD technique. The basic methodology is correct, the standard AMBER OL3 force field appears appropriate, as the first choice multipurpose RNA version. It is known to lead to over-compacted unstructured ssRNA ensembles, as all biomolecular force fields that are good for folded biopolymers. For the double strand, the circle and their flexibility it should be an optimal choice. The simulations are quite short by contemporary standards, though I do not think their prolongation would change the essence of the findings.

As noted above, I consider the experiments as very convincing. Strong point is that the accompanying simulations address a qualitative question and are clearly directed to solve it. It is a preferable application of the MD technique. So, I really like it, though I have some ideas for potential minor improvements, may be explanatory comments, all for supporting information.

There are some occasional typos, e.g. l. 180 stand displacement, l. 182 this suggest. I think on l. 56. where progress in non-enzymatic synthesis is overviewed, the reference could be more balanced. Appears to me that some groups are represented by duplicate citations while some research is omitted, for example a recent progress in template-free non-enzymatic RNA polymerization of 3',5' cyclic nucleotides , https://chemistry-europe.onlinelibrary.wiley.com/doi/10.1002/syst.202100017.

The short "jumping" supplementary movies are difficult to follow, I assume it is because of the size of the movies. Would it be possible to create a few SI Figures showing details of the most interesting parts of the structure, to focus on the key details, to accompany the movie?

In the simulations, lot of emphasis is paid on the Mg^2+^ simulations up to 500 mM MgCl. First point, is this condition relevant to the origin of life? Second, inclusion of divalents into MD is always risky, as they sample poorly (which is further exacerbated by the lack of bulk background due to the small periodic box, which may lead to glassy-like ion behavior around the solute). 400 ns is not sufficient to converge Mg^2+^. In addition, divalents, especially the high charge density Mg^2+^, are beyond the pair-additive MM approximation. It is impossible to simultaneously balance ion hydration and inner vs. outer shell binding to different coordination sites with the simple MM models. Could the authors briefly comment on initial placement of the ions after equilibration and during MD? Was it always hexacoordinated outer-shell binding to the RNA? Could the authors in SI briefly comment on it, and also comment if they had some specific reasons to choose the Duboue-Dijon parameters over parameters that have been more commonly used in biomolecular simulations? As I am not sure these specific parameters were tested/calibrated for RNA interactions (but I am not fully familiar with the work). Again, as the results are qualitative, I do not expect any effect on basic outcome of the work, so I am not suggesting any new computations.

---

## [Author Response]

Essential revisions:Reviewer #1:In the origin of life scenario where RNA is assumed to be the first replicator, a key problem is how RNA can replicate itself. Or how can RNA polymerase copy itself, since copying requires an open flexible structure that can be read. While the polymerase needs to be a topological rigid structure in order to catalyse the RNA polymer.The manuscript describes how a special kind of synthesis of the RNA, rolling circle synthesis on small pieces of circular RNA, could template and build RNA strands. The method apparently could help in avoiding the strand inhibition problem where stable RNA duplex in long strands hinders replication.It is a bit unclear how, but I suspect this i down to the size of the ring, since the problem increase with the length of the polymer. The authors also touch on this themselves in their MD simulations, since the entropically unfavourable confinement of a string onto a circle can alleviate part of the problem. Nevertheless, the authors also show by MD that the rings become small tight structures that actually hinders replication.

We welcome these comments. In the revised manuscript we have rewritten the relevant sections in order to clarify our arguments with respect to aspects of rolling circle synthesis that might aid RNA-catalyzed RNA replication.

In brief, our argument relies on the conjecture (supported by our MD simulations) that in small RNA rings (small circular RNAs, scRNAs), increasing strain upon RNA synthesis can contribute to the dissociation of double-stranded (ds) RNA into single-stranded (ss) RNA at both the 5’- or the 3’-RNA ends. Only the 3’- (but not the 5’-) RNA end is extended by the triplet polymerase ribozyme (TPR) and this process is irreversible. Therefore, over time there will be an overall shift of the dsRNA segment around the circle in the 3’- direction resulting in a 5’-ssRNA “tail” of increasing length. The efficiency of this process relies critically on the speed and processivity of the triplet polymerase ribozyme (TPR) (which is currently poor) and its ability stabilize the RNA 3’-end bound to the circRNA template and extend it before it can dissociate again.

Another potential issue is the fact that (as referee1 correctly observes), our MD simulations suggest RNA synthesis on scRNA templates stretches out the scRNA into an extended, rigid structure, which may be a poor template for replication. However, our experimental data suggest that rolling circle RNA synthesis is not only possible but can proceed over multiple full-length scRNA circles. We hypothesize that this reflects the fact that these rigid structures are likely in equilibrium with more relaxed structures, with more significant dissociation of the dsRNA segment from the circRNA template and / or kinks which relieve ring strain (even though these are not observed in our MD simulations) and it is those that can serve as templates for 3’-extension. Finally, although not supported by one-pot RCS experiments presented in Figure 4, it cannot be ruled out that the nascent strand comprises extension on two or even multiple scRNA templates. The precise mechanistic details do merit further study and this work is ongoing in our lab, but would lead too far for the current publication.

The study depends on trinucleotide triphosphates (triplets) as substrates. They demonstrate a viroid like replication that show how a template (-) is able to make a mirror copy (+) of circular RNA by a rolling circle synthesis with out the need for enzymes or other catalyst apart from RNA itself.While the conclusion of the work becomes a bit muddled, but honest, the work is a very important piece that demonstrates the huge potential role of circular RNA in the very early stages of life.

We thank the reviewer for his comments and have endeavored to make our conclusions as clear as possible. Furthermore, we have rewritten part of the Discussion section of the manuscript to increase clarity.

I must confess I am very far off my field of competences. I have tried my best to understand the paper, and the methods involved, but obviously I cannot give much feedback on the methods used. So, I cannot suggest improvements on that part.For the most part, the paper is well written, and the structure is sound. I would strongly recommend publication.It was unclear to me how it was certain that it was actually rings. There is the MD simulations, but apart from the principle drawing they are obviously not circular in shape.

Referee 1 raises an important point. Our study depends on the circularity of the RNA template molecules, i.e. small circular RNAs (scRNAs). We have therefore gone to some lengths to develop robust methods to efficiently generate such scRNAs, as well as to confirm their circularity experimentally. This is shown in Figure 1B and Supplementary Figure 1 (and described in Supplementary information Materials and Methods). scRNAs are generated by ligation from linear RNAs and circularity is confirmed by exonuclease degradation (exoT). Only scRNA can resist degradation by exoT, as only circular RNAs do not present a free RNA 3’-end as a substrate (Figure 1B). Circularity is further confirmed by electrophoretic mobility shift (Figure 1 —figure supplement 1 and Figure 4 —figure supplement 1).

To illustrate these points more clearly we have expanded the Figure legends for Figure 1B and Figure 1 —figure supplement 1, Figure 4 —figure supplement 1 and have also expanded the Materials and methods section describing experimental procedures to generate and purify scRNAs.

There are some parts of the manuscript that too me is very difficult to understand. Some statements seem to reflect a community understanding that might be obvious for those working with similar system on a day to day basis, but for the general reader (where I put myself) this becomes gibberish. for instance:RCS has potentially unique properties with regards to the strand inhibition problem where RNA duplex melting in principle can be effected by continuous toehold strand displacement driven by nucleotide hybridization and the ratchet of nascent strand extension by triphosphate hydrolysis. In an idealized RCS mechanism, such strand invasion and displacement processes are both isoenergetic and coordinated to nascent strand extension (Blanco et al., 1989; Daubendiek et al., 1995), with rotation of the single-stranded RNA (ssRNA) preventing the build-up of topological tension (Kuhn et al., 2002). Thus, RCS is a potentially open-ended process leading to the synthesis of single-stranded multiple repeat products (concatemers) with an internally energized strand displacement circumventing the "strand inhibition problem" (Tupper and Higgs, 2021).It might not be possible to rephrase this so that everyone can understand. But I think the clarity of manuscript could be improved.

We would like to apologize for failing to make our arguments sufficiently clear and free of jargon. We have now rephrased this passage in effort to make it both clearer as well as more accessible to the general reader. Please find below the rewritten section as found in the revised manuscript:

“Specifically in the context of triplet-based RNA replication on a circular template, duplex dissociation and strand separation may in principle be driven by trinucleotide (triplet) hybridization and ligation, leading to extension of the nascent strand 3’-end and an equal displacement of the 5’-end in triplet increments (Figure 1A). Triplet binding to the template strand and dissociation of an equal trinucleotide stretch from the 5’-end are both equilibrium processes and nearly isoenergetic. However, extension (i.e. ligation of the bound triplet to the growing 3’-end) is an irreversible step. Thus, in this scenario RCS would be expected to proceed in ratchet-like fashion with strand displacement driven by triphosphate hydrolysis and triplet ligation.”

But also because there is references to previous work where long statements does not make it more clear what is actually meant. for instance:Similar to what was described previously, RNA synthesis by the TPR best in the eutectic phase of water ice, due to beneficial reaction conditions for ribozyme catalysis such as reduced RNA hydrolysis and high ionic and RNA substrate concentrations (Attwater et al., 2010). This was also the case on scRNA templates.

Again, we would like to apologize for failing to make our arguments sufficiently clear to be easily understood. We have now rephrased this passage in an effort to make it both clearer as well as more accessible to the general reader. Please find below the rewritten section as found in the revised manuscript:

“As described previously, RNA synthesis by the TPR is most efficient in the eutectic phase of water ice, due its beneficial reaction conditions for ribozyme catalysis (Attwater et al., 2018). Specifically, eutectic ice phases aid TPR activity by the reduced degree of RNA hydrolysis under low temperature conditions, reduced water activity, and the high concentrations of reactants (ribozyme, scRNA template, triplet substrates and Mg^2+^ ions) present in the eutectic brine phase that arise by excluding solutes from growing ice crystals and remains liquid at subzero temperatures (Attwater et al., 2010). Thus, all RCS experiments were carried out under eutectic conditions."

That said, I think the overall message is clear and the paper is very interesting, I expect to reference it when it comes out.Reviewer #2:[…]Lines 122-126 – It is implied that triplets are better than monomers for rolling circle replication because triplets help to open up other double stranded regions. However, it is not obvious that this should be the case. To put a new triplet down you have to displace three bases from the displaced strand, whereas to put a monomer down you only have to displace one base. It is not easy to predict which of these is faster without measuring it. Furthermore, in the actual mechanism occurring here, there is no prior strand to be displaced at the point of attachment, because the displacement is occurring at the other side of the circle and it does not directly interfere with the attachment. So it is not clear whether this argument applies.

Referee 2 raises an important point and we welcome the opportunity to clarify our arguments.

The implication that triplets are able to “open up” double-stranded RNA regions is based on data in our previous publication (Attwater et al., 2018), which describes the properties of the triplet polymerase ribozyme (TPR) and the general TPR performance on linear RNA templates, which form stable secondary structures (such as hairpins). The key observation is that the TPR can replicate through such RNA structures as a function of triplet concentration, while the standard mononucleotide RNA polymerase ribozyme (RPR) cannot.

Referee 2 is of course correct in that triplets require the displacement of a 3 nt stretch from the primer / nascent strand 5’-end, whereas a mononucleotide RPR would require only a single base-pairing interaction to be disrupted. However, if this were the main energetic bottleneck of strand extension, surely the same argument would apply to template secondary structures, where again only one base-pairing interaction at the hairpin base would need to be disrupted compared to three for triplet invasion. However, our data show that only triplets endow the polymerase ribozyme with a general ability to invade and replicate through template secondary structures.

While we do not know the precise mechanistic basis for this, the fact that secondary structure invasion is a cooperative effect that occurs as a function of triplet concentration suggests, that it may be based on the ability of triplets to bind to intermittently accessible template structures. Triplet binding would stabilize them in their open form enabling the ribozyme to covalently link them to the nascent strand. Indeed, extension is an irreversible process (a ratchet), while 5’-end displacement is an equilibrium process suggesting that given sufficient time even an inefficient extension “ratchet” will “win”.

We of course accept that here may be other mechanisms for processive RNA synthesis on circular templates, such as e.g. tethering or topological linkage mechanism (as explored e.g. by (Cojocaru and Unrau, 2021)). However, we note that Cojocaru and Unrau do not investigate beyond full length circle synthesis and it is therefore currently unclear if their ribozyme is merely more processive on an “open“ template or if its superior activity extends to strand displacement.

We have now sought to explain our arguments more clearly in the revised manuscript (see also comments above).

Has replication of a circular strand actually been attempted with a monomer ribozyme? Is it known whether a triplet ribozyme is better than a monomer ribozyme on circular templates? If not, it would be better to avoid implying this.

We have not tested the mononucleotide RPR on circular RNA templates as – at the time this work was performed – all RPR variants required a hybridization tether to the template for processive RNA synthesis. On a circular RNA template this would likely cause both topological issues as well as – for full length synthesis – require displacement of the tether from the template for further extension. In contrast, the TPR does not require a tether for processive RNA synthesis by virtue of its non-catalytic t1 RNA subunit (Attwater et al., 2018) and this – together with above (2.1) described properties of triplets with regards to template secondary structure invasion – were the reasons we sought to explore triplet-based RCS. Recently, a more advanced RPR version has been described that appears to use a σ factor-like initiation mechanism that enables processive synthesis specifically on much larger, circular RNA templates (ca. 200 nt), where strain may be a lesser concern (Cojocaru and Unrau, 2021). However, as discussed above (2.1), strand displacement was not investigated in this report.

We have rewritten the relevant sections of the manuscript to clarify our arguments

Figure 1 – the periodic effect seen in 1E is claimed to be due to the difference of accessibility of template bases on the inside and outside of the circle. However, the results are measured by averaging over many different circular templates. I would expect that different copies of a circular template would have different configurations and would not always have the same bases on the inside. So the inside-outside difference should average out. Could the variability of 1E be explained by variation in rates of addition according to the sequence of the template rather than an inside-outside effect?

This is a perceptive observation and we have considered the exact same problem. Our explanation, even though speculative, is that the primer binds first and remains bound stably and therefore orients the nascent strand in a particular configuration with decreasing flexibility as the double-stranded RNA segment grows (as indicated by the MD simulations). Indeed, with primer bound and stably oriented – as illustrated in Figure 1F, we would expect to see the observed banding pattern based on accessibility of the triplet junctions. Again, the positioning of the primer is supported by the MD simulations (Figure 3A). As the template in this case is a UUC repeat, sequence variations (or variations in the rate of triplet addition according to the sequence of the template) are unlikely to account for the pattern observed. Indeed, we did not observe such a pattern in linear or non-repetitive circular template sequences. Furthermore, the same (or a very similar) periodic effect was observed on other repetitive circular templates, but not on their linear counterparts of identical sequence (see Figure 2 —figure supplement 1B).

The sequence effect would be the same in multiple copies of the same template sequence. Is there a similar variability seen when copying linear templates?

When copying linear templates (of similar sequence as the circularized RNA template) the periodic effect is not seen.

Figure 1C shows a TPR dimer. Is the polymerase actually in two parts? Is this important?

Yes, the triplet polymerase ribozyme holoenzyme is a heterodimer comprising an active (catalytic) RNA subunit (5TU) and an inactive subunit (t1). While the catalytic subunit by itself is a polymerase ribozyme, the non-catalytic subunit aids processive RNA synthesis (see Attwater et al., 2018). Therefore, the dimeric form of the TPR is the holoenzyme. This is important as the noncatalytic subunit is responsible for its ability to synthesize RNA in the absence of a template hybridization tether, which is specifically beneficial on circular RNA templates (as already outlined in more detail in 2.2). An in-depth description of the TPR and is properties can be found in our previous paper (Attwater et al. 2018). This has been specified in line 129-136 in the revised manuscript.

Line 213 – It is not clear why the 9 bp primer goes straight to 18 bp. What happened to the lengths in between?

In order to limit the number of MD runs, we concentrated on scRNAs with dsRNA segments reflecting later triplet additions (+3 triplets, 18 / + 4 triplets, 21 etc.) as we anticipated that ring strain would only begin to manifest itself in these molecules. Indeed, on scRNAs simulations with 9 bp and 18 bp of dsRNA, we observed an equally unbent double helical part, because the structure of the ssRNA part is longer and more relaxed and so it doesn’t exert any pulling force on the rest of the circle. We therefore did not model intermediate lengths 12 ( +1 triplet) and 15 (+ 2 triplets).

Line 220 – The word "extended" is used to mean that the unhybridized portion is stretched. There is a possible confusion with extending a strand by ligation of a triplet. Maybe the use of a word like stretched is better?

We thank the reviewer for this suggestion. “Extended” is indeed a questionable choice of word and liable to be confused with triplet extension. We have now replaced it with “stretched.”

Line 226 – The simulations show that the double-stranded part of the circle is stiff and only covers roughly half of the circle. The point at which the primer extension occurs is therefore far away from the point at which the two strands separate. This is important for very short circles. For longer circles, the stiffness should be less relevant, and there will come a point where the whole circle becomes double stranded. There will then need to be a true strand displacement occurring very close to the point of primer extension. How long would we need the circle to be before it switches to a double-stranded circle?

This is an important point. In the manuscript we suggest that this limit should be around the persistence length of RNA, which is about 200 bp. This mechanical property indicates the minimum length from which a polymer starts to behave flexibly, i.e. can significantly bend.

Does the stiffness effect seen here with the short circles make the primer extension reaction easier or more difficult than a true strand displacement reaction on a double stranded circle?

As discussed to some extent in (1.1) there are two conflicting mechanisms at work. On one hand, ring strain should aid strand displacement and therefore RNA synthesis beyond full length on a circular RNA template by destabilizing 5’-end hybridization and thereby aiding binding of triplets and extension. On the other hand, ring strain would be equally likely to destabilize 3’-end hybridization and thereby hinder extension and these two processes should therefore cancel each other out. Furthermore, there are other confounding factors such as restricted accessibility of some triplet junctions due to ring geometry (see Figure 1), which may further reduce extension efficiency compared to a linear RNA template. But 3’-end extension by covalent triplet ligation is a unidirectional, irreversible process and therefore overall we observe primer extension, rolling circle synthesis and strand displacement.

Lines 228-33 – This paragraph is not very clear. The meaning is not coming through.

We would like to apologize for not making the parsing clear and thank Reviewer 2 for making us aware of this. We have now reprised the paragraph to the following in effort to clarify:

“In the experimental data, we also observed an inhibitory effect for insertion of the final triplets (+8, +9, and +10 (beyond full length) / extension to 33, 36 and 39 nt of RNA in Figure 2D) into the corresponding scRNA template. This may indeed reflect the onset of the 3’- and 5’-end destabilization observed in the MD simulations (Figure 3), which would likely attenuate primer extension by the ribozyme. Note however that the extension efficiency recovered beyond full length (+11 / extension to 41 nt, Figure 2F), although at lower speed (Figure 2E).”

Line 288 – "orbit" is an odd word. Is there a better one?

We have now replaced “orbit” throughout the manuscript with “full length circle synthesis”. We agree with referee 2 that orbit may be liable to misinterpretation.

Figure 4 – Overall Figure 4 is not clear.– I have not understood the notation n: 3E5, 2E5 etc.– For sequence A Pos 4, GAA is the darkest shade, so I am presuming the template is CUU (in the reverse direction). But the second darkest shade is GGC. Why should GGC bind to CUU more strongly than others (for example GGA)? I am not sure whether I have understood this diagram correctly.

The diagram represents the deep sequencing result of a one-pot extension reaction where all four templates were mixed, in the presence of the TRP and triplets (CUG, AUA, CCA, CCC, AAA, CAC, GGG, UUA, UCC, GGC, AUC, GAU, CGC and GAA). The reads from the four templates were assorted relative to position 3 (the first barcode triplet). For the rest of the positions, the diagram presents (in %) which triplets were identified. The notation “n:” stands for number of reads used for the analysis at each position. 3E5, 2E5 represent the number (3*10^5, 2*10^5 etc.). This has now been specified in Figure 4B line 356-369 in revised manuscript.

The darkness of the shading represents the amount (%) of triplets found at the noted position. As correctly stated in the comments, the (reverse order) template for pos 4 is CUU and indeed GAA is the expected correct triplet. The expected triplet for each position is marked with a box in the diagram. In pos 4, the expected GAA is darkest shade. However, as he sequencing data indicated, GGC clearly is also misincorporated at this position to some degree and this is represented in the figure. This has been specified in the text line 328-341 in revised manuscript.

The overall fidelity of the TPR (per base) for linear templates is 97% (Atwater et al. 2018), but positional fidelity can vary and GC-rich triplets can be particularly prone to misincorporation both in a templated context (due to e.g. G-U mispairing) as well as in a non-templated context, as triplet terminal transferase extension of free unpaired 3’- ends.

– Part C shows fold difference. It would be easier if rates where shown for linear and circular strands separately. Why is sequence D a worse template? Or maybe it is not worse – it's just that the ratio of circular to linear templates is lower? It is not easy to understand this.

The fold difference in fidelity that is shown in Part C, is calculated based on the fidelity (%) presented in B, thus the ratio of templates should not play a role. However, to ensure that the concentration and integrity (circular or linear) of the templates were validated (by gel purification and absorbance measurement) and an equal amount of each were added in the experiment.

Calculation of plot in C is now specified in line 365-367 (in revised manuscript).

– Part D Figure 2 seems to show a double-stranded triplet being added. Why not just a single-stranded triplet.

A double stranded segment (consisting of e.g. two complementary triplets (triplet and anti-triplet, e.g. GGC and CCG), or a template and a number of triplets) will stack and may form a substrate for non-templated terminal transferase addition by the TPR. We have not discussed this in the manuscript as more data would be needed for a conclusive mechanistic description. As this remains therefore a somewhat speculative model (although consistent with the data obtained), we have now changed the figure to show a single stranded triplet as this is, as suggested, more intuitive. We have also added an explanation to the Figure 4 legend to the effect that the precise molecular nature of non-templated addition is currently unknown.

Figure 6D – It is unclear what is happening at each step. Particularly the backwards and forwards diagrams in step 4. Also, shouldn't the red strand be still attached to the blue circle before the cleavage occurs? The chemical structure in the middle is a bit distracting. I think the structure drawn in A is the same as D step 6. Maybe put parts A and D together and make B and C a separate figure?

We have changed the order in the figure so that the diagram with the reaction steps is presented before the data (see new Figure 6). In lines 424-430 (see revised manuscript) we have specified what the steps in the diagram denotes. Also, the Figure annotations for Figure 6A-D has been updated to fit the new Figure.

Line 436 – The reference to the virtual circular genome is misleading at this point. In the proposal of Zhou et al., there are no real circles, there are simply linear fragments that can be aligned to form a virtual circle. This does not fit with the rest of this paragraph. Either the reference to Zhou et al. should be omitted or it should be explained properly what the virtual circle proposal is.

We welcome this perceptive point and accept that our reference may be misleading in the context. However, while it is true that in the virtual circular genome proposal of Zhou et al., there are no real circles during the replication phase, Zhou et al. still propose circular RNAs arising from abiotic circularization reactions as an initial source of a circular genome. Nevertheless, we have now rewritten this section to clarify our arguments and removed the reference to Zhou et al.

Reviewer #3:Technically the experiments are sound, really comprehensive, convincing and the paper is well written. The documentation (the composed Figures etc.) is very nice. The MD simulations nicely complement the experiments. Strong point is that the simulations address a qualitative question and are clearly directed to solve it. It is a preferable application of the MD technique. The basic methodology is correct, the standard AMBER OL3 force field appears appropriate, as the first choice multipurpose RNA version. It is known to lead to over-compacted unstructured ssRNA ensembles, as all biomolecular force fields that are good for folded biopolymers. For the double strand, the circle and their flexibility it should be an optimal choice. The simulations are quite short by contemporary standards, though I do not think their prolongation would change the essence of the findings.As noted above, I consider the experiments as very convincing. Strong point is that the accompanying simulations address a qualitative question and are clearly directed to solve it. It is a preferable application of the MD technique. So, I really like it, though I have some ideas for potential minor improvements, may be explanatory comments, all for supporting information.There are some occasional typos, e.g. l. 180 stand displacement.

We have corrected this in the revised manuscript.

l. 182 this suggest. I think on l. 56. where progress in non-enzymatic synthesis is overviewed, the reference could be more balanced. Appears to me that some groups are represented by duplicate citations while some research is omitted, for example a recent progress in template-free non-enzymatic RNA polymerization of 3',5' cyclic nucleotides , https://chemistry-europe.onlinelibrary.wiley.com/doi/10.1002/syst.202100017.

We have focused our introduction and citations mainly on enzymatic (RNA-catalyzed) polymerisation of RNA and believe we provide a fair and balanced account and citation of the main advances in that specific field. Where we have briefly alluded to prebiotic chemistry including references to non-enzymatic RNA replication, we have concentrated on chemical approaches that are closely analogous to the enzymatic chemistry e.g. polymerization via an (activated) 5’phosphate. But reviewer 3 is of course correct that we should not discount alternative prebiotic mechanisms as these may have similarly given rise to early (circular) RNA templates for enzymatic replication. We have now therefore expanded and modified the citations in the introduction to include e.g. the above-mentioned publication as well as others.

The short "jumping" supplementary movies are difficult to follow, I assume it is because of the size of the movies. Would it be possible to create a few SI Figures showing details of the most interesting parts of the structure, to focus on the key details, to accompany the movie?

We thank the reviewer for his suggestion. We have created Figure 3 —figure supplement 1 and 2 with a series of snapshots along the two trajectories showing the main transition events.

In the simulations, lot of emphasis is paid on the Mg^2+^ simulations up to 500 mM MgCl. First point, is this condition relevant to the origin of life?

As now outlined in more detail in the manuscript, the RNA polymerization activity of our triplet polymerase ribozyme is enhanced in the eutectic phase of water ice at subzero temperatures. Such eutectic phases from when water containing solutes freeze and solutes such as Mg^2+^ ions, RNA etc. are excluded from the growing water-ice crystals into a liquid brine (eutectic) phase that surrounds the ice crystals and remains liquid at subzero temperatures. This process is accompanied by a profound concentration and dehydration effect, which enhances RNA synthesis and reduces RNA degradation, but also increases counter-ion concentrations to the above levels (0.5M).

The formation and presence of water ice on the early earth can of course not be proven, but is not implausible under currently favored geochemical scenarios for example at the poles or as part of seasonal or diurnal temperature variations. We have outlined our thinking on the beneficial properties of such ice phases in detail in one of the cited refs (Attwater et al., 2010) and now also discuss it in more detail within the main body of the revised manuscript. Furthermore, we now include an explicit reference to the rationale for simulations at high Mg^2+^ conditions with regards to the importance of eutectic ice phases in the experimental section.

Second, inclusion of divalents into MD is always risky, as they sample poorly (which is further exacerbated by the lack of bulk background due to the small periodic box, which may lead to glassy-like ion behavior around the solute). 400 ns is not sufficient to converge Mg^2+^. In addition, divalents, especially the high charge density Mg^2+^, are beyond the pair-additive MM approximation. It is impossible to simultaneously balance ion hydration and inner vs. outer shell binding to different coordination sites with the simple MM models. Could the authors briefly comment on initial placement of the ions after equilibration and during MD? Was it always hexacoordinated outer-shell binding to the RNA? Could the authors in SI briefly comment on it, and also comment if they had some specific reasons to choose the Duboue-Dijon parameters over parameters that have been more commonly used in biomolecular simulations? As I am not sure these specific parameters were tested/calibrated for RNA interactions (but I am not fully familiar with the work). Again, as the results are qualitative, I do not expect any effect on basic outcome of the work, so I am not suggesting any new computations.

We thank the reviewer for raising all these important points. We agree that Mg^2+^ represents a challenge for empirical force-field calculations due to strong polarization and charge-transfer effects. Indeed, nonpolarizable Mg^2+^ parameters tend to overestimate the formation of contact ion pairs by a factor of 2 to 3 (Fingerhut et al., 2021). We chose the parameters developed by Jungwirth and coworkers (Duboue-Dijon et al., 2018) because they drastically reduce ion−ion interactions with respect to ion−water interactions (see new Figure 3—figure supplement 6; Fingerhut et al; Kirby and Jungwirth, 2019). They account for the quantum effects by assuming a mean field approach using the so-called Electronic Continuum Correction (ECC), which is numerically implemented by rescaling the charge of multivalent ions, in the case of Mg^2+^ to 1.5.

We agree with the reviewer that 400 ns is short for an accurate description of the ion environment. However, we have added Figure 3 —figure supplement 6 showing reasonable convergence of ion location around RNA within the first and second hydration shell for the time course of our simulations.

We have added this extra information into the revised manuscript (Method section and Figure 3 – Supplement Figure 6).